# Systems-based identification of the Hippo pathway for promoting fibrotic mesenchymal differentiation in systemic sclerosis

Feiyang Ma [1,2,10], Pei-Suen Tsou [2,3,10], Mehrnaz Gharaee-Kermani[2], Olesya Plazyo[2], Xianying Xing[2], Joseph Kirma[2], Rachael Wasikowski [2], Grace A. Hile[2], Paul W. Harms [1,4], Yanyun Jiang[1], Enze Xing [2], Mio Nakamura[1], Danielle Ochocki[2,3], William D. Brodie[2], Shiv Pillai [5], Emanual Maverakis[6], Matteo Pellegrini [7], Robert L. Modlin [7,8], John Varga[2,3], Lam C. Tsoi[1], Robert Lafyatis [9], J. Michelle Kahlenberg [2], Allison C. Billi [1], Dinesh Khanna [2,3] ✉ & Johann E. Gudjonsson [1] ✉

Systemic sclerosis (SSc) is a devastating autoimmune disease characterized by excessive production and accumulation of extracellular matrix, leading to fibrosis of skin and other internal organs. However, the main cellular participants in SSc skin fibrosis remain incompletely understood. Here using differentiation trajectories at a single cell level, we demonstrate a dual source of extracellular matrix deposition in SSc skin from both myofibroblasts and endothelial-to-mesenchymal-transitioning cells (EndoMT). We further define a central role of Hippo pathway effectors in differentiation and homeostasis of myofibroblast and EndoMT, respectively, and show that myofibroblasts and EndoMTs function as central communication hubs that drive key pro-fibrotic signaling pathways in SSc. Together, our data help characterize myofibroblast differentiation and EndoMT phenotypes in SSc skin, and hint that modulation of the Hippo pathway may contribute in reversing the pro-fibrotic phenotypes in myofibroblasts and EndoMTs.

Systemic sclerosis (SSc, scleroderma) is a devastating autoimmune-driven connective tissue disease associated with mortality rates that in its most severe form are comparable to metastatic cancers[1,2]. The etiology of SSc remains poorly understood but involves genetic predisposition[3], female sex[4], and environmental exposures[5,6], resulting in autoimmunity, fibrosis in the skin and internal organs, along with prominent microvasculopathy[7]. Current treatment paradigms can slow, but not halt or reverse, fibrosis[8].

Skin thickening in SSc is due to increased deposition of extracellular matrix (ECM) components, most prominently type I collagen[9,10]. With disease progression, there is increasing dermal fibrosis and loss of adnexa[11]. The increased skin tightness correlates with accumulation of myofibroblasts in the skin[12,13]. Myofibroblasts are mesenchymal cells of fibroblast lineage that are transiently activated during normal wound healing[14]. In contrast to fibroblasts, myofibroblasts express the contractile protein alpha smooth muscle actin, and

[1]Department of Dermatology, University of Michigan, Ann Arbor, MI, USA. [2]Division of Rheumatology, Dept of Internal Medicine, University of Michigan, Ann Arbor, MI, USA. [3]University of Michigan Scleroderma Program, Ann Arbor, MI, USA. [4]Department of Pathology, University of Michigan, Ann Arbor, MI, USA. [5]Ragon Institute, Massachusetts General Hospital, Massachusetts Institute of Technology and Harvard University, Boston, MA, USA. [6]Department of Dermatology, University of California, Davis, Sacramento, CA, USA. [7]Dept of Molecular, Cell, and Developmental Biology, University of California Los Angeles, Los Angeles, CA, USA. [8]Division of Dermatology, Department of Medicine, University of California, Los Angeles, CA 90095, USA. [9]Division of Rheumatology, University of Pittsburgh, Pittsburgh, PA, USA. [10]These authors contributed equally: Feiyang Ma, Pei-Suen Tsou. ✉e-mail: khannad@med.umich.edu; johanng@med.umich.edu

form characteristic stress fibers[15]. Myofibroblasts are commonly first detected in the deep dermis[16], and their persistence in SSc[17,18] is thought to be responsible for the exaggerated and uncontrolled ECM production[15,18]. The factors driving differentiation and accumulation of myofibroblasts in SSc have been extensively investigated. It is currently thought that TGF-β and mechanical forces are key factors in myofibroblast development[14,19]. Notably, mechanical stiffness itself activated latent TGF-β[20–23], setting up a self-sustaining amplification circuit for tissue fibrosis in SSc skin[10].

Vascular dysfunction and structural abnormalities, particularly involving the arterioles, are among the earliest manifestations of SSc and precede development of fibrosis[24]. This vascular dysfunction results in altered capillary architecture with reduced vessel density, a hallmark of SSc, causing decreased capillary blood flow and tissue hypoxia[25]. Clinically, these microvascular changes underlie painful digital ulcerations, pulmonary arterial hypertension, gastric vascular ectasia, mucocutaneous telangiectasia and scleroderma renal crisis[26–28]. Larger vessels, including arterioles in the lung and the kidney, may show intimal proliferation (onion skinning) and adventitial fibrosis, accompanied by loss of pericytes. Apoptosis of endothelial cells in SSc was first noted over 25 years ago[29], and there is accumulating evidence that this endothelial damage may be immune-mediated[30]. Immune mechanisms underlying the endothelial damage in SSc include cytotoxic T cells[8] and anti-endothelial autoantibodies[31]. Upon microvascular injury, endothelial cells can acquire a mesenchymal phenotype through a process termed endothelial-to-mesenchymal transition (EndoMT)[32]. EndoMT is characterized by the loss of cell-cell junctions and endothelial markers, such as von Willebrand factor, CD31, and VE-cadherin, coupled with the acquisition of invasive properties and gain of mesenchymal markers such as alpha SMA and collagens[33]. EndoMT has been implicated in driving the pathogenesis of fibrosis in multiple fibrotic conditions, including SSc[34].

The Hippo pathway is a highly conserved signaling pathway that regulates cell proliferation, apoptosis, and stemness in response to a wide range of extracellular and intracellular signals. Downstream effects of Hippo pathway signaling are mediated by the YAP and TAZ transcription co-activators through binding with members of the TEAD family of transcription factors[35]. When the Hippo pathway is inactive, YAP/TAZ enters the nucleus, competes with VGLL family of transcription co-factors for binding to TEADs, and recruits other factors to induce gene transcription[35,36]. When Hippo pathway is active, YAP/TAZ is phosphorylated by LATS1/2 on multiple sites and is retained in the cytoplasm due to interaction with 14-3-3 proteins and eventually removed through poly-ubiquitination and degradation[35]. Hippo signaling has been shown to regulate the expression of ligands for WNT, TGF-β, JAK-STAT, EGFR, and Notch pathways[37], placing this pathway at the nexus of multiple biological processes, many of which have been implicated in SSc pathogenesis[38–42].

Previous studies using single cell RNA-sequencing (scRNA-seq) of SSc skin[10,30,43–45], focused on particular aspects of SSc pathogenesis including the heterogeneity of fibroblasts[44,46] and myofibroblasts[10], endothelial cell injury[30], and monocyte[43], and T cell responses[45]. However, these studies either lacked spatial context or incompletely explored cellular trajectories and the signaling mechanisms involved in myofibroblast differentiation and vascular dysfunction that are characteristic of SSc. Here, we sought to provide a detailed and comprehensive characterization of the cellular and molecular events underlying skin fibrosis using SSc skin biopsies from a large cohort of patients.

Here, we provide a detailed and comprehensive characterization of the cellular and molecular events underlying skin fibrosis using SSc skin biopsies from a large cohort of patients. We identify the Hippo pathway as a potential therapeutic target in systemic sclerosis, and describe its role in both myofibroblast and EndoMT transition, two key cellular events in SSc pathogenesis, and provide a link with the sex-biased Hippo pathway regulator VGLL3.

## Results

### scRNA-seq and spatial-seq reveal diverse cell types and their spatial locations in SSc skin

To understand the unbiased cellular composition and cell states of healthy normal skin (NS) and systemic sclerosis (SSc) skin, we generated single cell suspensions of skin biopsies from 18 NS donors and 22 SSc patients and performed scRNA-seq (Supplementary Data 1). The resulting quality-controlled SSc plus NS single cell atlas contained a total of 96,174 cells, with an average of 2258 genes and 9263 transcripts detected per cell. The cells were clustered based on differential expression of marker genes and visualized on a uniform manifold approximation and projection (UMAP) plot. Cluster annotation was corroborated by overlapping the cluster markers with the canonical lineage-specific genes reported in previous skin disease scRNA-seq studies[10,30,43–45]. We recovered 12 major cell types across all the samples (Fig. 1a–d, Supplementary Data 2), including keratinocytes, melanocytes, eccrine gland cells, endothelial cells, fibroblasts, pericytes, smooth muscle cells, nerve cells, T cells, myeloid cells, mast cells, and B cells. Most of these cell types contained cells from both NS and SSc samples (Fig. 1b, c). Notably, two small populations, smooth muscle cells, and B cells, were primarily derived from the SSc samples (Fig. 1c). Interestingly, clear separations were observed in keratinocytes, fibroblasts, pericytes, and endothelial cells between the NS and SSc cells, suggesting fundamental transcriptional differences between them (Fig. 1b). By contrast, the other cell types displayed overlapping patterns for the NS and SSc cells.

To localize the major cell types detected by scRNA-seq in SSc skin, we performed spatial sequencing (spatial-seq) on four independent SSc skin samples using the 10X Visium platform. After the quality control steps, we detected 951 spatially defined spots with an average of 805 genes and 1647 transcripts per spot (Supplementary Fig. 1a). We deconvoluted the spatial spots by the major cell types detected in scRNA-seq using the Seurat anchor-based label transfer method (Methods). The deconvolution scores for each cell type were displayed on the tissue (Fig. 1e) and combined into a scatter-pie plot representing the relative cell type composition for each spot (Supplementary Fig. 1a). As expected, keratinocytes localized to the epidermis and the hair follicle. Melanocytes were enriched along the basal epidermis. Myeloid cells and T cells were primarily located in the superficial dermis in proximity to the epidermis and the hair follicle. B cells were aggregated adjacent to the hair follicle. Strikingly, fibroblasts, pericytes, and endothelial cells were distributed throughout a large proportion of the spots in the dermis, covering the regions of fibrosis. The other cell types represent small populations and were lowly detected in the spatial-seq sample. Spatial-seq of three additional SSc skin biopsy specimens analyzed in parallel revealed similar results (Supplementary Fig. 1b–d).

### Hippo pathway effectors promote and maintain myofibroblast differentiation in SSc skin

Due to their capacity to produce large amounts of pro-fibrotic extracellular matrix (ECM) components, myofibroblasts have long been regarded as central to the fibrotic response in SSc. A recent scRNA-seq study proposed *SFRP2*hi fibroblasts as the potential progenitors of myofibroblasts in SSc skin[10]. To understand the heterogeneity of the fibroblasts and the myofibroblast differentiation process in SSc skin, we selected all the fibroblasts from our scRNA-seq dataset and performed sub-clustering (Supplementary Fig. 2a). Based on previously published marker genes[10,47–49], we annotated the fibroblast subclusters into seven subtypes including *SFRP2*+ fibroblasts (FB), *COL8A1*+ FB, *CCL19*+ FB, *FMO1*+ FB, *FMO2*+ FB, *TNN*+ FB, and *CLDN1*+ FB (Fig. 2a, e, Supplementary Data 2). The *COL8A1*+ FB expressed high

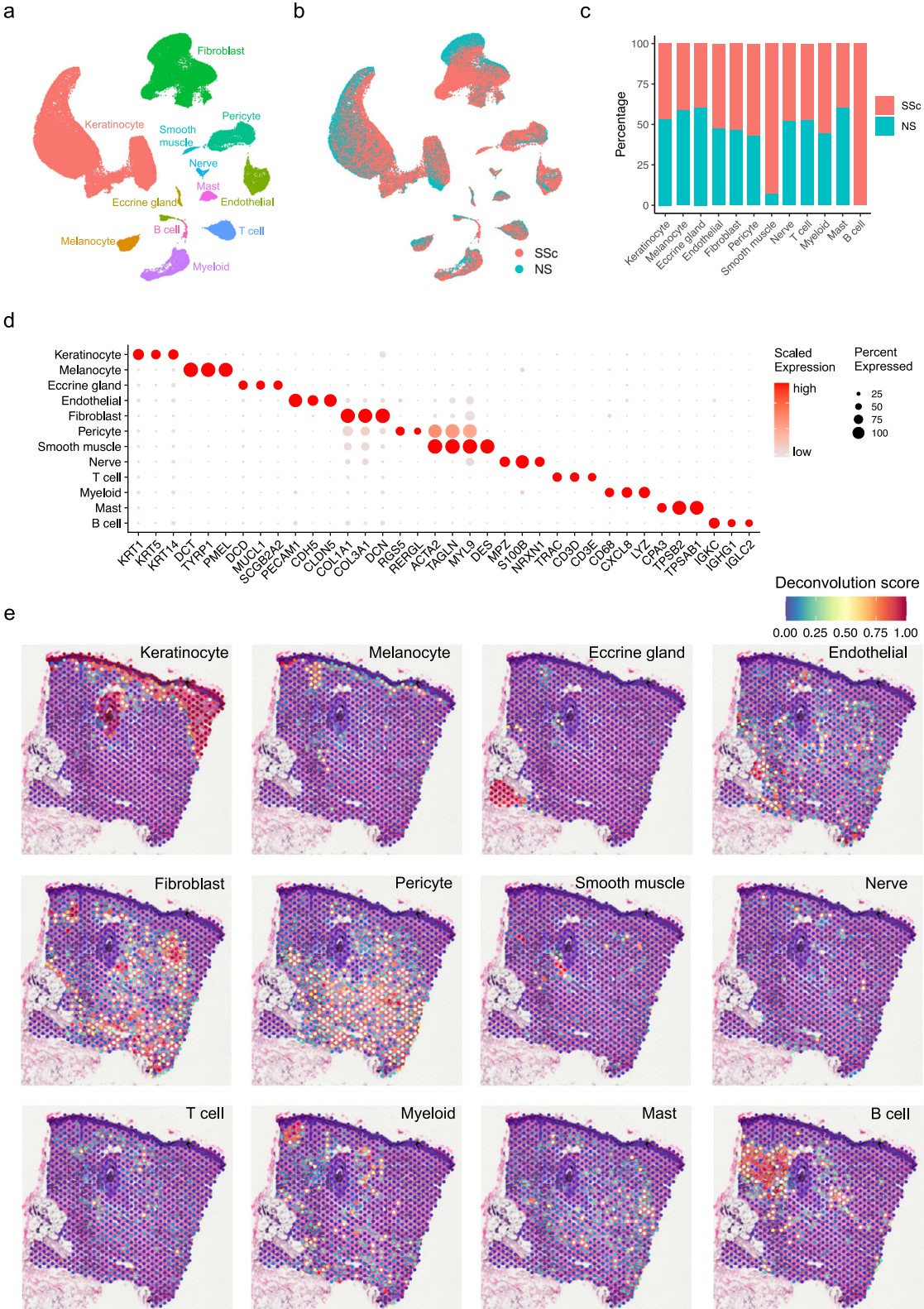

**Fig. 1 | Cell types observed in SSc skin and their spatial locations. a** UMAP plot showing 96,174 cells colored by cell types. **b** UMAP plot showing the cells colored by disease conditions. SSc systemic sclerosis, NS normal skin. **c** Bar plot showing the abundance composition across the disease conditions for each cell type in scRNA-seq. **d** Dot plot showing representative marker genes for each cell type. The color scale represents the scaled expression of each gene. The size of the dot represents the percentage of cells expressing each gene of interest. **e** Spatial plot showing the deconvolution score for each cell type. The coordinates of the spot correspond to the location in the tissue (spatial data representative of *n* = 4).

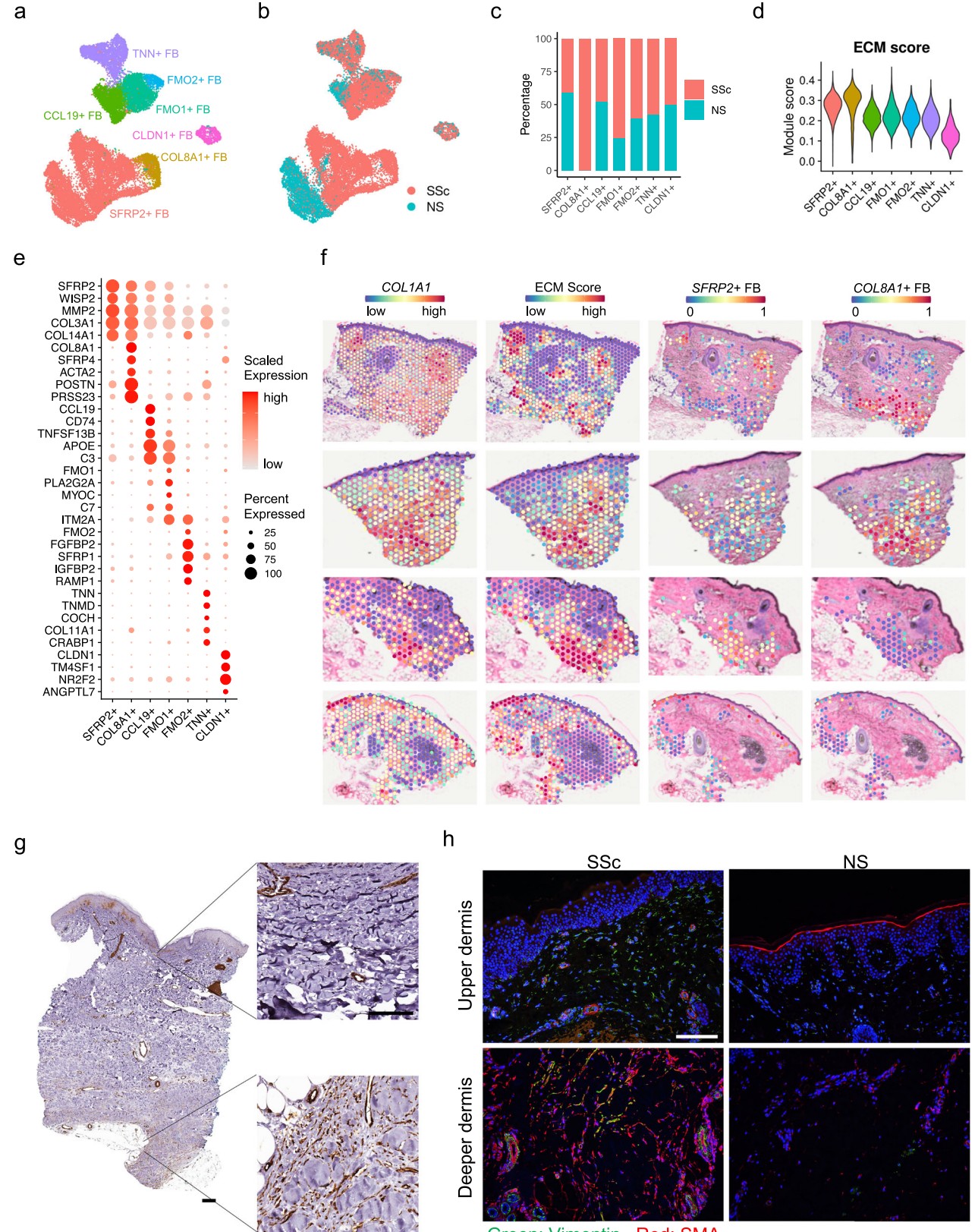

levels of *ACTA2*, *SFRP4*, *POSTN*, and *PRSS23*, representing myofibroblasts in our dataset[10]. Strikingly, the *COL8A1*+ FBs, which were derived exclusively from the SSc samples, were contiguous with the *SFRP2*+ FB on the UMAP (Fig. 2b, c). Notably, the NS and SSc *SFRP2*+ FB were largely distinct on the UMAP, suggesting fundamental transcriptional differences between these two cell populations. Furthermore, only the

SSc *SFRP2*+ FB, not the NS ones, connected to the *COL8A1*+ FB (Fig. 2a, b), suggesting that only in SSc did *SFRP2*+ FB have the potential of directly differentiate into myofibroblasts. To illustrate the capacity for ECM production by the different fibroblast subtypes, we calculated an ECM module score using the gene list from the extracellular matrix pathway from Gene Ontology (Fig. 2d) and plotted all the collagen

**Fig. 2 | Identification of fibroblast subtypes and their spatial locations. a** UMAP plot showing 25,182 fibroblasts colored by subtypes. **b** UMAP plot showing the fibroblasts colored by disease conditions. **c** Bar plot showing the abundance composition across the disease conditions for each fibroblast subtype. **d** Violin plot showing the extracellular matrix module scores in the fibroblast subtypes. **e** Dot plot showing the top marker genes for each fibroblast subtype. The color scale represents the scaled expression of each gene. The size of the dot represents the percentage of cells expressing the gene of interest. **f** The left two panels show the *COL1A1* expression and extracellular matrix module score across all the spots in four spatial-seq samples. The right two columns show the deconvolution score for the *SFRP2*⁺ FB and *COL8A1*⁺ FB in the fibroblast-rich spots. **g** Immunohistochemistry staining for SMA in SSc skin tissue. The size bar represents 100 μm (staining is representative of $n = 3$). **h** Immunofluorescence showing the colocalization of Vimentin and SMA in the SSc and NS tissues (staining is representative of $n = 3$). The size bar represents 100 μm.

genes across the subtypes (Supplementary Fig. 2b). The *COL8A1*⁺ FB had the highest ECM score and highest expression pattern of the collagen genes, including *COL1A1*, *COL1A2*, *COL3A1*, *COL5A1*, *COL5A2*, *COL6A3*, *COL8A1*, *COL8A2*, *COL10A1*, and *COL12A1*. Not surprisingly, the *SFRP2*⁺ FB ranked the second highest for the ECM score and collagen gene expression. To corroborate the spatial locations of the fibroblasts, we next plotted the expression of *COL1A1* and the ECM score across the four spatial-seq samples and found them highly correlated with the fibroblast deconvolution scores (Figs. 1e, 2f, S1a–d). We then deconvoluted the fibroblast-rich spots (with fibroblast deconvolution score >0.25) using the fibroblast subtypes (Fig. 2f, S2c). The *SFRP2*⁺ FB were primarily located in the superficial dermis, while the *COL8A1*⁺ FB were localized to the deeper dermis. We did not detect *COL8A1*⁺ FB in the third and fourth spatial-seq samples, likely due to sample heterogeneity and that we did not cut deep enough in these two tissue sections. Immunohistochemistry and immunofluorescence of SMA (encoded by *ACTA2*) validated the location of myofibroblasts in the deeper dermis of SSc skin (Fig. 2g, h), overlapping with areas of intense fibrosis. Interestingly, SMA from the upper dermis of the SSc skin tissue was observed mainly in and around the blood vessels (Fig. 2g, h), suggesting endothelial to mesenchymal transition (EndoMT) in SSc skin.

To better characterize the differentiation process from *SFRP2*⁺ FB to *COL8A1*⁺ FB, we focused on these two subtypes and split them into three groups based on the fibroblast sub-clusters (Fig. 3a, S2a). Group 1 and 2 were mainly composed of NS *SFRP2*⁺ FB and SSc *SFRP2*⁺ FB, respectively, and group 3 was composed of *COL8A1*⁺ FB (Fig. 3b). As expected, the expression of myofibroblast marker genes *ACTA2*, *TAGLN*, and *COL8A1* and the ECM score increased from group 1 to 2 to 3 (Fig. 3c, e). We calculated module scores using genes induced in cultured fibroblasts stimulated by two pro-fibrotic cytokines, TGF-β or IL-4, and found that these two module scores escalated from group 1 to 2 to 3 (Fig. 3e, Supplementary Data 3, Methods). Interestingly, the expression of two Hippo pathway target genes, *CTGF* and *CYR61*, displayed increasing trends from group 1 to 2 to 3 (Fig. 3f), suggesting involvement of the Hippo pathway in myofibroblast differentiation in SSc skin. Furthermore, CytoTRACE analysis[50] suggested that group 1 represented less differentiated cells and group 3 represented the most differentiated cells (Fig. 3d). Based on these observations, we performed pseudotime analysis on the three groups using Monocle and arranged the cells into a linear trajectory from group 1 to 2 to 3 (Fig. 3g, h). To identify the potential cytokines and transcription factors that drive the myofibroblast differentiation, we split the variable genes along the pseudotime into five expression patterns (Supplementary Fig. 3a, Supplementary Data 4). We then inferred the upstream regulators for genes in each pattern using Ingenuity Pathway Analysis. For each upstream regulator, we calculated a module score using all target genes gathered from the five expression patterns (Methods). Consistent with the cytokine stimulation experiment, both TGF-β1 and IL-4 target scores were highly correlated with the pseudotime (Fig. 3i). Previously reported transcription factors, STAT1, IRF7, RUNX1 and SMAD3, that promote the differentiation from *SFRP2*ʰⁱ fibroblast to myofibroblasts[10], displayed a module score positively correlated with the pseudotime (Fig. 3i). The module scores for WWTR1 (TAZ) and TEAD1 were also highly correlated with the

pseudotime, suggesting a contribution of the Hippo pathway in the differentiation from the *SFRP2*⁺ FB to the *COL8A1*⁺ FB. Indeed, multiple key molecules involved in Hippo pathway, including *WWTR1*, *CTGF*, *CYR61*, *TEAD1*, *TEAD2*, *TEAD3*, *TEAD4*, and *VGLL3*, were highly expressed in group 3 compared to groups 1 and 2 (Supplementary Fig. 3b).

To determine the functional relevance of the Hippo pathway in SSc fibroblasts, we used primary dermal fibroblasts isolated from patients with early diffuse cutaneous (dc)SSc (Supplementary Data 1). Two inhibitors relevant to the Hippo pathway were used; TRULI, a LATS1/2 (large tumor suppressor kinase 1 and 2) inhibitor that decreases phosphorylation of YAP to promote its entry into the nucleus[51], and verteporfin, an inhibitor of YAP-TEAD interaction[52]. In all assays, the two inhibitors showed opposite effects. TRULI treatment led to a significant increase in pro-fibrotic markers in dcSSc fibroblasts including *ACTA2* and *COL1A1* (Fig. 3j), and we validated this for SMA with western blotting, but the increase in COL1A1 did not reach significance (Fig. 3k). Further, these findings were validated by immunofluorescent staining with increased expression of SMA but only slight to moderate increase in COL1A1 expression (Fig. 3l). Specifically for verteporfin, we found that its treatment in dcSSc fibroblasts downregulated both SMA and COL1 to levels comparable to what was observed in healthy fibroblasts at baseline (Supplementary Fig. 3c, d). Since myofibroblasts exhibit higher contractility, increased proliferation, and migration capacities, we also measured these as functional endpoints. We found that TRULI enhanced gel contraction, proliferation, and migration in dcSSc fibroblasts (Fig. 3m–o), suggesting that TRULI promotes a pro-fibrotic phenotype in these cells. In contrast, verteporfin downregulated SMA and COL1 in dcSSc fibroblasts at both mRNA levels and protein levels (Fig. 3j–l). Verteporfin also inhibited gel contraction, proliferation, and migration in a dose-dependent manner (Fig. 3m–o). We also determined the impact of these inhibitors in healthy dermal fibroblasts. Verteporfin did not appear to have any effect on fibrotic gene expressions, proliferation, and migration in healthy fibroblasts, while TRULI upregulated SMA, though to a lesser extent compared to dcSSc fibroblasts, it did not affect proliferation or migration in healthy fibroblasts (Supplementary Fig. 3e, f). To further delineate the roles of key mediators in the Hippo pathway in SSc fibroblasts, we knocked down *YAP1*, *TEAD1*, *TEAD3*, *VGLL3*, or *TEAD1*/*TEAD3* simultaneously in these cells (Fig. 3p) and measured *ACTA2* and *COL1A1* levels. Similar to what was observed with verteporfin treatment, knockdown of these key mediators resulted in downregulation of *ACTA2* and *COL1A1* (Fig. 3q). Altogether, these results point to a role for Hippo pathway effectors in promoting and maintaining the pro-fibrotic signal in dcSSc fibroblasts.

## Hippo pathway effectors promote and maintain endothelial to mesenchymal transition in SSc skin

To characterize the heterogeneity of endothelial cells, we subclustered all the endothelial cells from the scRNA-seq dataset and obtained seven sub-clusters (Fig. 4a–c, Supplementary Data 2). Sub-cluster 6 represented the lymphatic endothelial cells with high expression of *LYVE1* and *MMRN1* (Fig. 4c). Sub-clusters 3, 4, and 5 represented the activated endothelial cells with high expression of *SELP* and *SELE* (Supplementary Fig. 4a). Interestingly, sub-clusters 0, 1, and 2, expressed *GJA4*, a marker of arteriolar endothelial cells[53], and

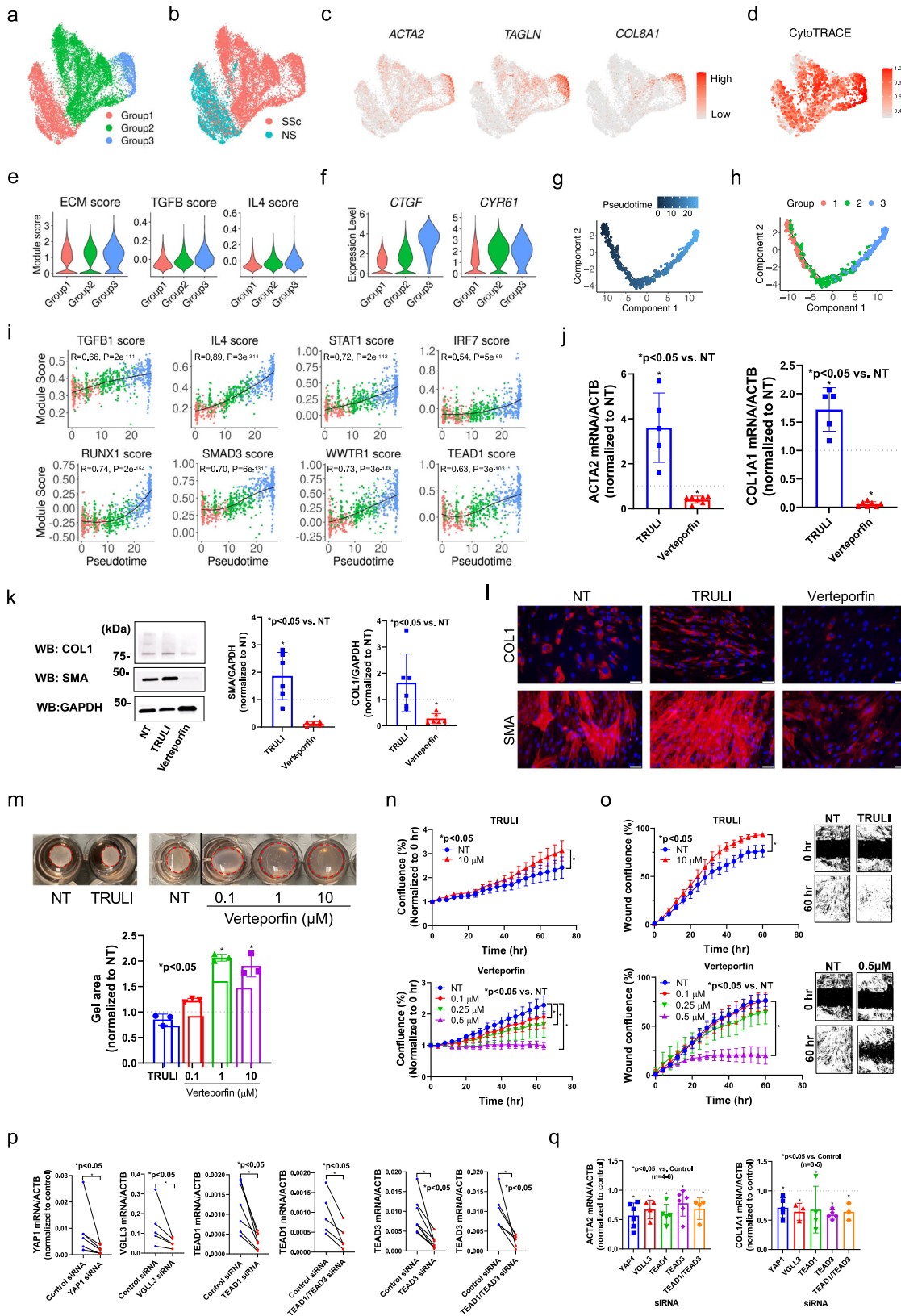

were aligned contiguously on the UMAP, suggesting a potential transition among these sub-clusters. Close examination of lineage marker genes in the three sub-clusters reinforced this transition hypothesis: expression of endothelial cell markers *PECAM1* and *CDH5*, as well as Notch signaling genes *NOTCH1*, *JAG2*, and *DLL4*, decreased, while mesenchymal cell markers *COL1A1*, *ACTA2*, and *TAGLN* increased from

sub-cluster 0 to 1 to 2 (Fig. 4d). Immunofluorescence confirmed co-expression of SMA and CD31 (encoded by *PECAM1*) in SSc but not NS skin (Fig. 4g). Sub-cluster 2 also had the highest ECM score (Fig. 4e) and expressed the highest level of collagen genes among all the endothelial sub-clusters (Supplementary Fig. 4b). The disease composition revealed increasing proportions of SSc endothelial cells from sub-

**Fig. 3 | Hippo pathway regulates myofibroblast differentiation in SSc skin.**
**a** UMAP plot showing the *SFRP2*⁺ FB and *COL8A1*⁺ FB colored by groups. **b** UMAP plot showing the *SFRP2*⁺ FB and *COL8A1*⁺ FB colored by disease conditions. **c** UMAP plots showing the expression level of *ACTA2*, *TAGLN* and *COL8A1* in the three fibroblast groups. **d** UMAP plot showing the CytoTRACE score in the three fibroblast groups. A higher CytoTRACE score suggests the cell being more differentiated. **e** Violin plots showing the extracellular matrix, TGF-β and IL-4 module scores in the three fibroblast groups. **f** Violin plots showing the expression level of *CTGF* and *CYR61* in the three fibroblast groups. **g** Pseudotime trajectory colored by the pseudotime of the three fibroblast groups. **h** Pseudotime trajectory colored by the group identity of the three fibroblast groups. **i** Scatter plots showing the correlation between the fibroblast pseudotime and the target score or the upstream regulators. The color represents the group identity of the cell. Correlation test was applied. **j** Quantitative PCR results showing the effect of TRULI or verteporfin (both 10 μM) on *ACTA2* and *COL1A1* expression in dcSSc fibroblasts. Data normalized to NT. $N = 5–7$ patient lines. Data presented as mean +/− SD. Mann–Whitney test was applied. $P < 0.05$ was designated as statistically significant. $p = 0.0079$ for the TRULI groups, $p = 0.0006$ for the verteporfin groups. **k** Effect of TRULI or verteporfin (both 10 μM) on COL1 and SMA levels by Western blotting. Quantification of samples were from different blots, however the blots were processed in parallel and the data of each patient line is normalized to its own NT group. $N = 5–6$ patient lines. Data presented as mean +/− SD. Two-sided unpaired *t*-test was applied. $P < 0.05$ was designated as statistically significant. $p = 0.035$ for SMA-TRULI, $p = 0.0001$ for SMA-verteporfin, $p = 0.0001$ for COL1-verteporfin. **l** Immunofluorescence showing TRULI enhanced while verteporfin Inhibited COL1 and SMA expression in dcSSc

fibroblasts. The size bar represents 50 μm. **m** TRULI enhanced while verteporfin blocked gel contraction in dcSSc fibroblasts. Data normalized to the corresponding NT group. $N = 3$ patient lines. Data presented as mean +/− SD. Kruskal–Wallis test or Mann–Whitney test was applied for verteporfin or Truli, respectively. $P < 0.05$ was designated as statistically significant. $p = 0.019$ and $p = 0.049$ for verteporfin 1 μM and 10 μM. **n** TRULI increased cell proliferation while verteporfin dose-dependently blocked cell growth. Cell proliferation was monitored by analyzing the occupied area by cells over time, using the IncuCyte S3 Analysis software. $N = 3$ patient lines. Data presented as mean +/− SEM. Two-way ANOVA test was applied. $P < 0.05$ was designated as statistically significant. $P = 0.0001$ for all significant groups. **o** TRULI enhanced cell migration while verteporfin blocked migration in a dose-dependent manner. Two-way ANOVA test was applied. $N = 4$ patient lines. Data presented as mean +/− SEM. $P < 0.05$ was designated as statistically significant. $P = 0.0001$ for all significant groups. **p** Extent of knockdown of genes relevant in the Hippo pathway in dcSSc fibroblasts. $N = 5$ patient lines. Two-sided paired *t*-test or Wilcoxon test was applied. $P < 0.05$ was designated as statistically significant. *YAP1* siRNA: $p = 0.016$; *TEAD1* siRNA: $p = 0.0037$; *TEAD1/TEAD3* siRNA: $p = 0.017$ for *TEAD1*, $p = 0.016$ for *TEAD3*; *TEAD3* siRNA: $p = 0.0078$. **q** Knockdown of genes involved in the Hippo pathway resulted in downregulation of *ACTA2* and *COL1A1*. $N = 4–6$ patient lines. Two-sided unpaired *t*-test was applied. $P < 0.05$ was designated as statistically significant. Data presented as mean +/− SD. *ACTA2*: $p = 0.0006$ for *YAP1* and *TEAD1* siRNA; $p = 0.0059$ for *VGLL3* siRNA; $p = 0.031$ for *TEAD3* siRNA; $p = 0.014$ for *TEAD1/TEAD3* siRNA. *COL1A1*: $p = 0.0051$ for *YAP1* siRNA; $p = 0.013$ for *VGLL3* siRNA; $p = 0.0001$ for *TEAD3* siRNA, $p = 0.013$ for *TEAD1/TEAD3* siRNA. Source data is provided for this figure.

cluster 0 to 1 to 2 (Supplementary Fig. 4c). Furthermore, the Cyto-TRACE score indicated that the cells in sub-cluster 2 were more differentiated compared to the cells in subcluster 0 and 1 (Fig. 4f). Taken together, the above evidence suggests EndoMT occurs during progression from sub-cluster 0 to 1 to 2.

To better characterize the EndoMT progression at the single cell level, we performed pseudotime analysis on these three sub-clusters using Monocle and aligned them into a linear trajectory (Fig. 4h). The pseudotime was assigned from early to late from sub-cluster 0 to 1 to 2. In parallel to the fibroblast trajectory analysis, we calculated the correlation coefficient between the target score for different upstream regulators and the pseudotime in the endothelial trajectory analysis (Supplementary Fig. 4d). We focused on the key upstream regulators that promote the *SFRP2*⁺ FB to the *COL8A1*⁺ FB transition and plotted the correlation coefficients in both fibroblast and endothelial trajectory analyses (Fig. 4i). As expected, TGF-β transcription factors displayed positive correlation in both the fibroblast and the endothelial transitions. Intriguingly, TEAD transcription factors were also highly correlated with both transitions, suggesting that Hippo pathway effectors promote both myofibroblast transition and EndoMT. Accordingly, key molecules involved in Hippo pathway, including *YAP1, WWTR1, CTGF, CYR61, TEAD1, TEAD2, TEAD3, TEAD4*, and *VGLL3*, were all expressed in higher percentage of cells in sub-cluster 2 compared to sub-clusters 0 and 1 (Fig. 4k). By contrast, the target scores of HIF1A, STAT1, IRF7, and RUNX1 were only positively correlated with the fibroblast pseudotime but not the endothelial pseudotime (Fig. 4i). We validated the expression of TEAD1 and TEAD3 by immunofluorescence and demonstrated their colocalization with CD31 in SSc but not NS samples (Fig. 4j). Given the similarity between fibroblast transition and endothelial transition, we sought to determine the common programs that were upregulated in both processes. To do so, we overlapped the up-regulated genes in group 3 compared to group 1 and 2 in the fibroblast transition and those in sub-cluster 2 compared to sub-cluster 0 and 1 in the endothelial transition. Enrichment analysis on the 240 common up-regulated genes implicated the extracellular matrix organization as the most prominent program commonly induced in both transitions (Fig. 4l, Supplementary Data 5).

We next determined whether the Hippo pathway is functionally involved in EndoMT in endothelial cells isolated from dcSSc skin. TRULI increased *ACTA2* and *COL1A1* expression in dcSSc endothelial

cells while verteporfin downregulated these genes (Fig. 5a). Interestingly, TRULI had inconsistent effects on endothelial markers *PECAM1* and *CDH5*, while verteporfin blocked the expression of both at the mRNA levels. The effect of these inhibitors on dcSSc endothelial cells were further validated using Western blotting (Fig. 5b). The expression levels in SSc ECs treated with verteporfin appeared to be comparable to the expression levels in ECs isolated from healthy controls. Of note, we were unable to detect COL1 expression in healthy ECs. By immunofluorescent staining we showed that TRULI enhanced EndoMT as shown by cell morphology and loss of endothelial marker VWF. In contrast, verteporfin reversed EndoMT by downregulating SMA expression in dcSSc endothelial cells (Fig. 5c). We also determined the effects of these inhibitors on ECs isolated from healthy donors. It appears that TRULI induced EndoMT by downregulating VWF, upregulating SMA, and also induced morphological changes, while verteporfin appeared to increase VWF (Fig. 5c). The discrepancy of the effect of the inhibitors on EC markers in Fig. 5a–c could be due to posttranscriptional or translational modifications, or differences in protein and RNA degradation rates. To further dissect the roles of YAP1, TEAD3, or VGLL3 in mediating EndoMT in dcSSc endothelial cells, we knocked these genes down individually (Fig. 5d) and measured EndoMT markers. Knockdown of *YAP1, TEAD3*, or *VGLL3*, resulted in lower *ACTA2* and *COL1A1* levels, similar to what was observed with verteporfin treatment (Fig. 5e). TEAD3 and VGLL3 knock-down appeared to promote endothelial markers *PECAM1* and *CDH5*, while *YAP1* knock-down had minimal effect. The differences that we observed between verteporfin and the knockdown studies on vascular gene expressions in Fig. 5a, e could be due to the off-target effects of verteporfin. We also speculate that TEAD3 and VGLL3 might be playing more critical roles in the Hippo Pathway as opposed to YAP1 alone. The individual and combined contributions of these mediators in EndoMT in SSc require further investigation. Nevertheless, these results suggest that Hippo pathway effectors play a significant role in EndoMT in dcSSc endothelial cells.

## Myofibroblasts and EndoMTs act as central hubs in cell-cell communications

To comprehensively study the cell type composition and cell-cell communications in SSc skin, we sub-clustered the other cell types in the scRNA-seq dataset. We identified four subtypes of keratinocytes:

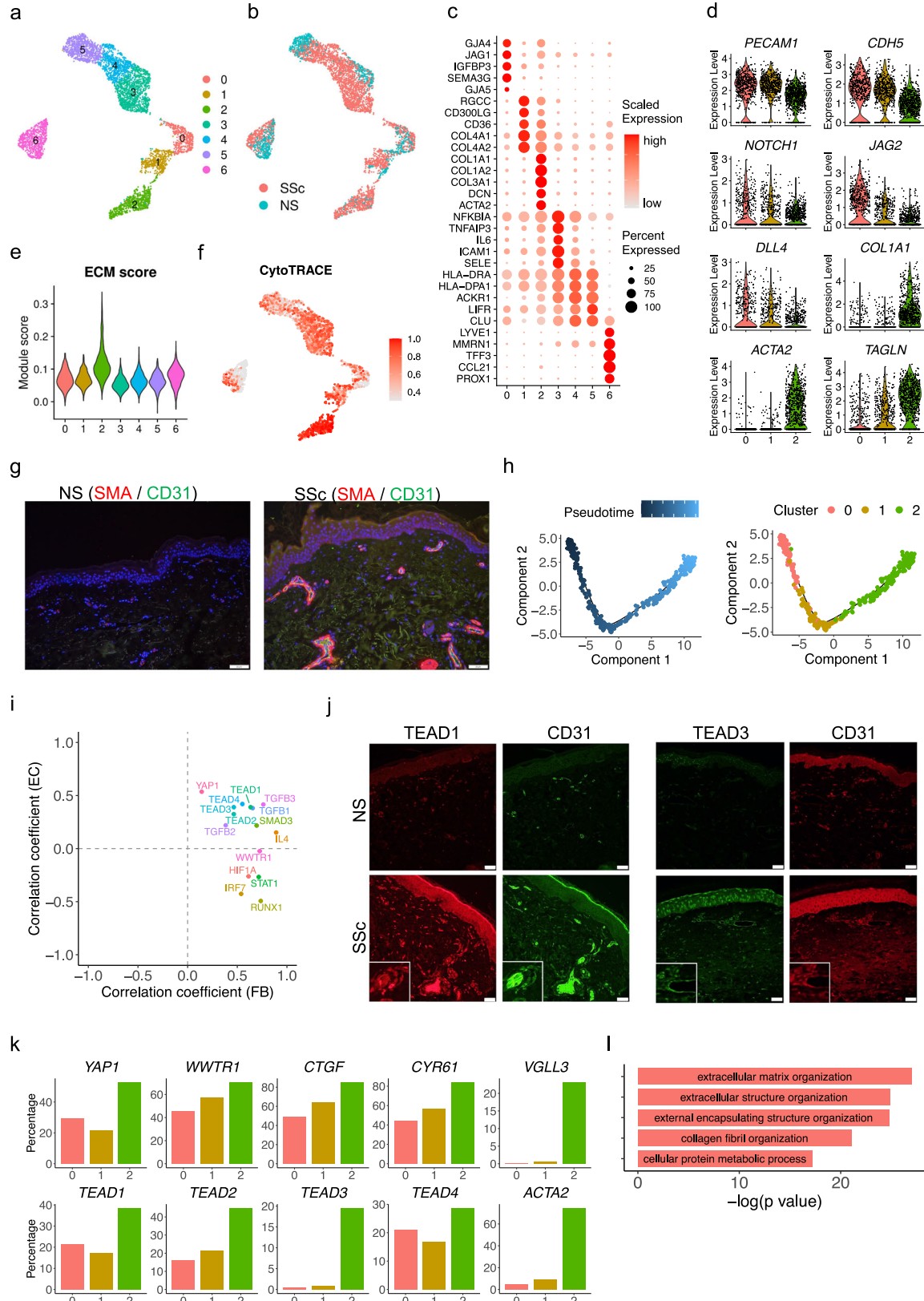

basal, spinous, supraspinous, and inflammatory keratinocyte (Supplementary Fig. 5a). The T cells were sub-clustered and annotated into five subtypes: CD4 T cell (CD4T), regulatory T cell (Treg), CD8 T cell (CD8T), nature killer cell (NK cell), and innate lymphoid cell (ILC) (Supplementary Fig. 5b). For myeloid cells, we annotated six subtypes, including Langerhans cell (LC), conventional type 1 DC-like cell (cDC1), conventional type 2 A DC-like cell (cDC2A), conventional type 2B DC-like cell (cDC2B), monocyte (Mono), and macrophage (Mac) (Supplementary Fig. 5c). We sub-clustered the pericytes and smooth muscle cells together due to their transcriptional similarities. We obtained four pericyte sub-clusters and two smooth muscle cell sub-clusters (Supplementary Fig. 5d). For the other cell types, we obtained five

**Fig. 4 | Characterization of endothelial to mesenchymal transition in SSc skin.** **a** UMAP plot showing 5070 endothelial cells colored by sub-clusters. **b** UMAP plot showing the endothelial cells colored by disease conditions. **c** Dot plot showing the top marker genes for each endothelial sub-cluster. The color scale represents the scaled expression of each gene. The size of the dot represents the percentage of cells expressing the gene of interest. **d** Violin plots showing the expression level of representative genes in the endothelial sub-cluster 0, 1 and 2. **e** Violin plots showing the extracellular matrix module score in the endothelial sub-clusters. **f** UMAP plot showing the CytoTRACE score in the endothelial sub-clusters. A higher CytoTRACE score suggests the cell being more differentiated. **g** Immunofluorescence showing the colocalization of SMA and CD31 in the NS and SSc skin tissues. Images shown are representative of n = 3. The size bar represents 20 μm. **h** Pseudotime trajectory colored by the pseudotime (left) and sub-cluster identity (right) of three endo-thelial sub-clusters. **i** Scatter plot showing the correlation coefficients between the target score of the upstream regulators and the fibroblast pseudotime (X axis) and endothelial pseudotime (Y axis). **j** Immunofluorescence showing the colocalization of TEAD1/CD31 (left) and TEAD3/CD31 (right) in the NS and SSc skin tissues. Images shown are representative of n = 3. The size bar represents 20 μm. **k** Bar plots showing the percentage of cells expressing the gene in the endothelial sub-cluster 0, 1 and 2. **l** Bar plot showing the top five Gene Ontology pathways enriched for the 240 common up-regulated genes in fibroblast group 3 compared to group 1, 2 and endothelial sub-cluster 2 compared to sub-cluster 0, 1.

eccrine gland cell sub-clusters, six melanocyte sub-clusters, four nerve cell sub-clusters, five mast cell sub-clusters, and three B cell sub-clusters (Supplementary Fig. 6a–e).

We then performed separate ligand-receptor analyses on the SSc and NS cellular subtypes using CellPhoneDB (Methods). To assess the changes that occur in SSc, we analyzed ligand-receptor pairs from the subtypes that had higher interaction scores in SSc compared to NS. By aggregating the total number of interactions for each cell type, we observed the largest ligand-receptor pair number changes in endo-thelial cells and fibroblasts (Fig. 6a). Connectome web analysis revealed endothelial sub-cluster 2 (EC2, EndoMTs) and the *COL8A1*[+] FB (myofibroblasts) as the central communication hubs. Of note, EC2 and *COL8A1*[+] FBs had the greatest number of self-interactions (Supple-mentary Fig. 7a), many of which overlapped between EC2 and *COL8A1*[+] FB cells as these two subtypes share fibrotic features. Thus, these overlapping self-interactions account for some of the interactions highlighted between the two groups. Taken together, these data sug-gest dynamic interaction between these two key subtypes and other skin cell subtypes during the EndoMT and fibroblast to myofibroblast transition (Fig. 6c). Of note, several other endothelial and fibroblast subtypes were also relatively more active, including EC0, EC1, EC3, EC4, *SFRP2*[+] FB, *FMO1*[+] FB, and *CCL19*[+] FB, which mainly interacted with the two hubs. By contrast, fewer ligand-receptor interactions were higher in NS compared to SSc, with the majority of these representing keratinocyte-keratinocyte crosstalk (Supplementary Fig. 7b, c). To this end, we focused on the interactions within the SSc endothelial cells and fibroblasts. To emphasize the roles of the two hubs, we selected the pairs from which the ligands were subtype-specific marker genes for either EC2 or *COL8A1*[+] FB, which uncovered various validated and uncharacterized signaling pathways implicated in fibrosis (Fig. 6c, Supplementary Data 6). Fibroblast growth factors (*FGF2*, *FGF7*, and *FGF18*)[54], platelet-derived growth factor (*PDGFA* and *PDGFC*)[55,56], transforming growth factor beta (*TGFB1* and *TGFB3*), and vascular endothelial growth factor (*VEGFA* and *VEGFB*)[57] were known to be involved in fibrosis. *WNT2* and *WNT4* were specifically expressed by the *COL8A1*[+] FB and have been reported to promote cardiac fibrosis by activating NF-κB signaling[58]. *IL6* and *IL11* served as pro-fibrotic cyto-kines; both have been reported to be elevated in SSc and proposed as therapeutic targets for SSc[59,60]. Notably, our analysis revealed several other pro-inflammatory mediators, including *CCL2*, *CCL8*, *CCL11*, *TNFSF4*, *TNFSF9*, and *TNFSF12*, that were highly expressed by EC2 or the *COL8A1*[+] FB. Together, these data illustrate pro-fibrotic shifts within the interactome in SSc and reinforce the essential roles of EC2 and the *COL8A1*[+] FB in SSc skin fibrosis.

## Discussion

Fibroblast-myofibroblast transition and vascular dysfunction have long been recognized as key pathologic features in SSc[17,18,24]. Vascular abnormalities involving the arterioles are among the earliest mani-festations occurring in almost every SSc patient[25]. This vascular dys-function precedes development of fibrosis and manifests clinically as Raynaud's phenomenon, digital ulcerations, and increased promi-nence of nailfold capillary loops[24,25]. The subsequent fibrosis is thought

to result from exaggerated and uncontrolled production of ECM components mainly by myofibroblasts in SSc skin[15,18]. However, the relationship between the vascular dysfunction and triggering of fibrosis has remained unclear.

The data presented here reinforces the premise that myofibro-blasts are derived from the *SFRP2*[+] FB subset[10]. In the SSc environment, the *SFRP2*[+] FB progressively acquires qualities of myofibroblasts as it transitions into fully developed myofibroblasts defined by *ACTA2* and *COL8A1* expression. The *SFRP2*[+] FB and myofibroblasts are the major source of *COL1A1* expression by fibroblastic lineage cells in SSc skin. Strikingly, with spatial-seq and scRNA-seq integration, we observe prominent compartmentalization of the fibrotic process in SSc skin, with increased *COL1A1* expression and ECM activity occurring in localized areas both superficially and deep in the dermis. Those areas correspond to the *SFRP2*[+] FB and myofibroblast signatures, with the *SFRP2*[+] FB being most prominent in the superficial dermis whereas myofibroblasts predominate in the deeper layers of the dermis, as previously described[16].

These data further demonstrate that EndoMT represents another source of ECM in SSc skin, which provides further insight into the molecular mechanisms driving EndoMT at single cell resolution. The EndoMT trajectory started amongst a cluster of endothelial cells expressing *GJA4*, encoding connexin37. Connexin37 is found in arter-iolar endothelial cells, including afferent but not efferent arterioles in the kidney[61,62], and has been shown to be regulated by Notch signaling leading to arrest of endothelial growth and promotion of arterial specification[53]. Notably, connexin37-dependent mechanisms have been shown to modulate angiotensin II-mediated hypertension[63], a pathway that has been implicated in scleroderma renal crisis[64]. Our findings of EndoMT in arteriolar and pre-capillary endothelial cells align with the reported vascular dysfunction in SSc[24]. Intriguingly, *GJA4* has been reported to be more highly expressed in female tissue[65], but whether its expression and EndoMT are influenced by the sex-biased transcription co-factor VGLL3 will need to be determined in future studies.

Notably, most of the cellular interactions activated in SSc skin involved only two major cell types, endothelial cells, and fibroblasts. Of those, the most pronounced cell-cell interactions were seen between the EndoMTs (EC2) and myofibroblasts (*COL8A1*[+] FB). These interac-tions involve different chemokines (*CCL2*, *CCL8*, *CCL11*, *CXCL1*), cyto-kines (*IL6*, *IL11*, *TGFB1*, *TFGB3*, *TNFSF4*, *TNFSF9*, *TNFSF12*), growth factors (*FGF2*, *FGF7*, *FGF18*, *VEGFA*, *VEGFB*), and WNT ligands (*WNT2*, *WNT4*). Many of these mediators have been implicated in SSc patho-genesis including *CCL2* and *CCL8* in promoting dendritic cell differentiation[66], *IL6* and *IL11* in promoting fibrosis[39,59,67], *TNFSF4* as genetic predisposition in SSc[68], *FGF7* in fibroblast activation[69], *VEGF* in vascular dysfunction[70], and *WNT2* in dermal fibrillin deposition[40]. Despite previously reported interactions, many ligand-receptor inter-actions inferred from gene expression require experimental validation in future studies.

Myofibroblasts are commonly observed in wound healing and are thought to be terminally differentiated cells that undergo apoptosis after wound contraction, as they are rarely found in non-pathological

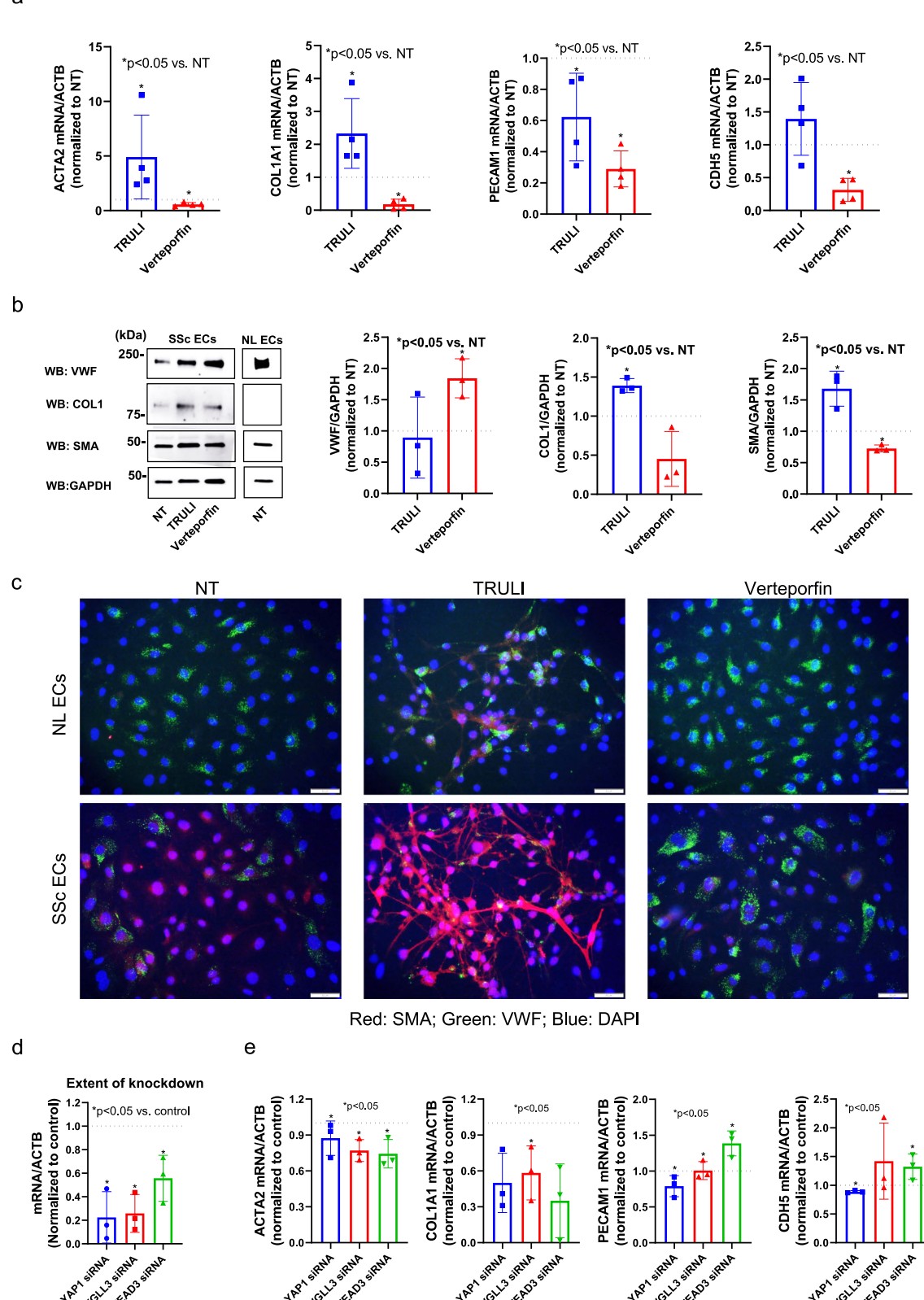

Red: SMA; Green: VWF; Blue: DAPI

situation[71]. Members of the Hippo pathway have been implicated in myofibroblast differentiation[72,73] and fibrosis[74], with the TEAD inhibitor verteporfin reducing kidney fibrosis in a mouse model[75] and pro-fibrotic phenotype in SSc skin-derived fibroblasts[76]. YAP is kept out of the nucleus through intact LATs1/2 activity under normal physiologic states, whereas knock-out of LATS induces spontaneous fibroblast to myofibroblast differentiation[73]. YAP and TEAD largely exert their transcriptional activity by interacting with distal enhancers and regulate transcription via multiple mechanisms including recruitment of general transcription factors, modification of epigenetic markers, and modulation of chromatin structure[35,77–79]. Multiple soluble factors and cytokines regulate the Hippo pathway. These include mediators

**Fig. 5 | Hippo pathway regulates endothelial to mesenchymal transition in SSc skin. a** The effect of TRULI (10 μM) and verteporfin (1 μM) on *ACTA2, COL1A1, PECAM1,* and *CDH5* expression in dcSSc endothelial cells. Data normalized to NT. *N* = 4 patient lines. Data presented as mean +/− SD. Mann–Whitney test was applied. *P* < 0.05 was designated as statistically significant. *P* = 0.029 for all the groups that were marked significant. **b** Western blotting showing the effect of TRULI or verteporfin on VWF, COL1, and SMA in dcSSc endothelial cells. The expression levels of each protein in healthy dermal ECs are shown for comparison. Quantification of samples were from different blots, however, the blots were processed in parallel and the data of each patient line is normalized to its own NT group. *N* = 3 patient lines. Two-sided unpaired *t*-test was applied. *P* < 0.05 was designated as statistically significant. *P* = 0.0095 for VWF-verteporfin, *p* = 0.0016 for COL1-TRULI, *p* = 0.014 for SMA-TRULI, *p* = 0.001 for SMA-verteporfin. NL normal. **c** Immunofluorescence showing TRULI enhanced the mesenchymal phenotype while verteporfin promoted the endothelial phenotype in dcSSc endothelial cells, while in healthy ECs, TRULI induced EndoMT to a lesser extent, while verteporfin had minimal effect. Images shown are representative of *N* = 3 patient lines. Scale bar = 50 μm. **d** The extent of knockdown of *YAP1, VGLL3,* or *TEAD3* in dcSSc endothelial cells. *N* = 3 patient lines. Data normalized to control and presented as mean +/− SD. Two-sided unpaired *t*-test was applied. *P* < 0.05 was designated as statistically significant. *P* = 0.0036 for *YAP1* siRNA, *p* = 0.0013 for *VGLL3* siRNA, *p* = 0.017 for *TEAD3* siRNA. **e** Knockdown of genes involved in the Hippo pathway blocked the EndoMT phenotype in dcSSc endothelial cells. Two-sided unpaired *t*-test was applied. *P* < 0.05 was designated as statistically significant. Data presented as mean +/− SD. *N* = 3 patient lines. *ACTA2: p* = 0.012 for *VGLL3* siRNA, *p* = 0.02 for *TEAD3* siRNA; *COL1A1: p* = 0.025 for *YAP1* siRNA, *p* = 0.033 for *VGLL3* siRNA, *p* = 0.022 for *TEAD3* siRNA; *PECAM1: p* = 0.018 for *TEAD3* siRNA; *CDH5: p* = 0.0007 for *YAP1* siRNA. Source data is provided for this figure.

signaling through various G-protein coupled receptors, the Wnt/Beta-catenin pathway[80], EGF[81], BMP, and the TGF-β pathway[82,83]. However, what triggers increased trafficking of effectors of Hippo pathway to the nucleus of SSc *SFRP2*⁺ FB and activates the fibroblast to myofibroblast differentiation remains unclear and will be the focus of future studies.

A recent publication by Gur et al. described a novel *LGR5*⁺ fibroblast subset in SSc pathogenesis and highlighted its involvement in signaling pathways implicated in SSc[46]. While in the present studies, we were able to detect *LGR5* expression in SSc and normal skin fibroblasts, we were unable to confirm that these cells represented a distinct subpopulation, either in our own data or that of others[10]. We found that *LGR5* was detected in only a minority of fibroblasts (17% of the *SFRP2*⁺ FB, 12% of the myofibroblasts, and <5% of the other fibroblast subpopulations). The reasons for these differences in the frequency of *LGR5*⁺ fibroblasts between these two studies are currently unclear, but may relate to differences in the depth of RNA sequencing and the use of more sensitive library preparation method in our cohort.

Treatment of SSc fibroblasts with the LATS inhibitor TRULI, which enhances YAP nuclear localization and transcriptional activity, promoted myofibroblast development and increased collagen production, whereas the inverse was observed with TEAD inhibition by verteporfin. The same pattern was observed for endothelial cells, as TRULI induced a mesenchymal phenotype while verteporfin reversed it in SSc skin-derived endothelial cells. The mechanisms involved are likely very complex in both myofibroblast differentiation and EndoMT. The TEAD family consists of four members (TEAD1-4). In addition to interacting with YAP and TAZ, TEAD1-4 can interact with VGLL family of proteins (VGLL1-4)[84]. The VGLL-binding site overlaps with the YAP/TAZ binding site on TEADs[85]. When VGLLs and YAP/TAZ are both present in the nucleus, they likely compete for TEAD binding[85] and, therefore, differentially modulate TEAD transcriptional effects. VGLL3 has been previously implicated as a promoter of sex-biased autoimmune responses[86], and VGLL3 has recently been implicated in cardiac fibrosis promoting myofibroblast collagen production[87]. Our data are consistent with such a role for VGLL3 both in relation to myofibroblast function and collagen I expression, as well as development of EndoMTs, raising the intriguing possibility that VGLL3 may contribute to the female bias in SSc through its modulation on Hippo-TEAD signaling in myofibroblasts or EndoMTs, or both. To address the role of each one of the TEADs and VGLLs in mediating myofibroblast differentiation and EndoMT, more work is required, including addressing the role of VGLL3.

The data presented here also implicate non-stromal cell populations in SSc pathogenesis, particularly immune cells including B cells and T cells. B cells were highly enriched in SSc skin and may provide a link between pathological processes in the skin and development of autoantibodies, found in most patients with SSc[88]. CD8 T cells have been postulated to serve as key drivers in the prominent endothelial dysfunction in SSc[8], with some of the earliest features SSc being

perivascular edema along with perivascular mononuclear infiltrate particularly in the upper and mid-dermis[11]. These early changes coincide with the same dermal locations where EndoMTs, characterized by SMA + CD31+, are prominent. Besides immune cells, other stromal cells, such as the smooth muscle cells, which were primarily derived from the SSc biopsies, could also play a role in pathogenesis. The immune-stromal cell crosstalk is outside the focus of the current study, but future work will be required to further address and characterize those interactions.

In summary, our study provides an unprecedentedly detailed view of the cellular composition and cell-cell communications in SSc skin. We demonstrate striking compartmentalization of fibrotic processes in skin and identify two major profibrotic cellular subtypes, myofibroblasts and EndoMTs, as the principal cell types responsible for fibrosis in SSc skin. We further highlight the central role of Hippo signaling in promoting development of both SSc myofibroblasts and EndoMTs (Supplementary Fig. 8). Lastly, we provide preclinical evidence that the profibrotic phenotypes of myofibroblasts and EndoMTs in SSc can be reversed by inhibiting TEAD transcriptional activity, a finding that will need to be tested and validated in future studies and ultimately, in clinical trials.

## Methods

### Human sample acquisition
22 systemic sclerosis patients and 18 healthy donors were recruited for single cell RNA sequencing, and additional 4 systemic sclerosis patients were recruited for spatial sequencing. Skin biopsies were taken from the affected forearm of patients. The study was approved by the University of Michigan Institutional Review Board (IRB), and all patients gave written consent. The study was conducted according to the Declaration of Helsinki Principles.

### Single-cell RNA-seq library preparation, sequencing, and alignment
Generation of single-cell suspensions for scRNA-seq was performed as follows: Skin biopsies were incubated overnight in 0.4% dispase (Life Technologies) in Hank's Balanced Saline Solution (Gibco) at 4 °C. Epidermis and dermis were separated. Epidermis was digested in 0.25% Trypsin-EDTA (Gibco) with 10 U/mL DNase I (Thermo Scientific) for 1 h at 37 °C, quenched with FBS (Atlanta Biologicals), and strained through a 70 μM mesh. Dermis was minced, digested in 0.2% Collagenase II (Life Technologies) and 0.2% Collagenase V (Sigma) in plain medium for 1.5 h at 37 °C and strained through a 70 μM mesh. For the samples collected from the University of Pittsburgh, epidermal and dermal cells were combined in 1:1 ratio. For three of the samples collected from the University of Michigan, the epidermal and dermal cells were prepared in different libraries that were constructed by the University of Michigan Advanced Genomics Core on the 10X Chromium system with chemistry v3. For the remaining samples, the epidermis and dermis were combined in 1:1 ratio. Libraries were then sequenced on the

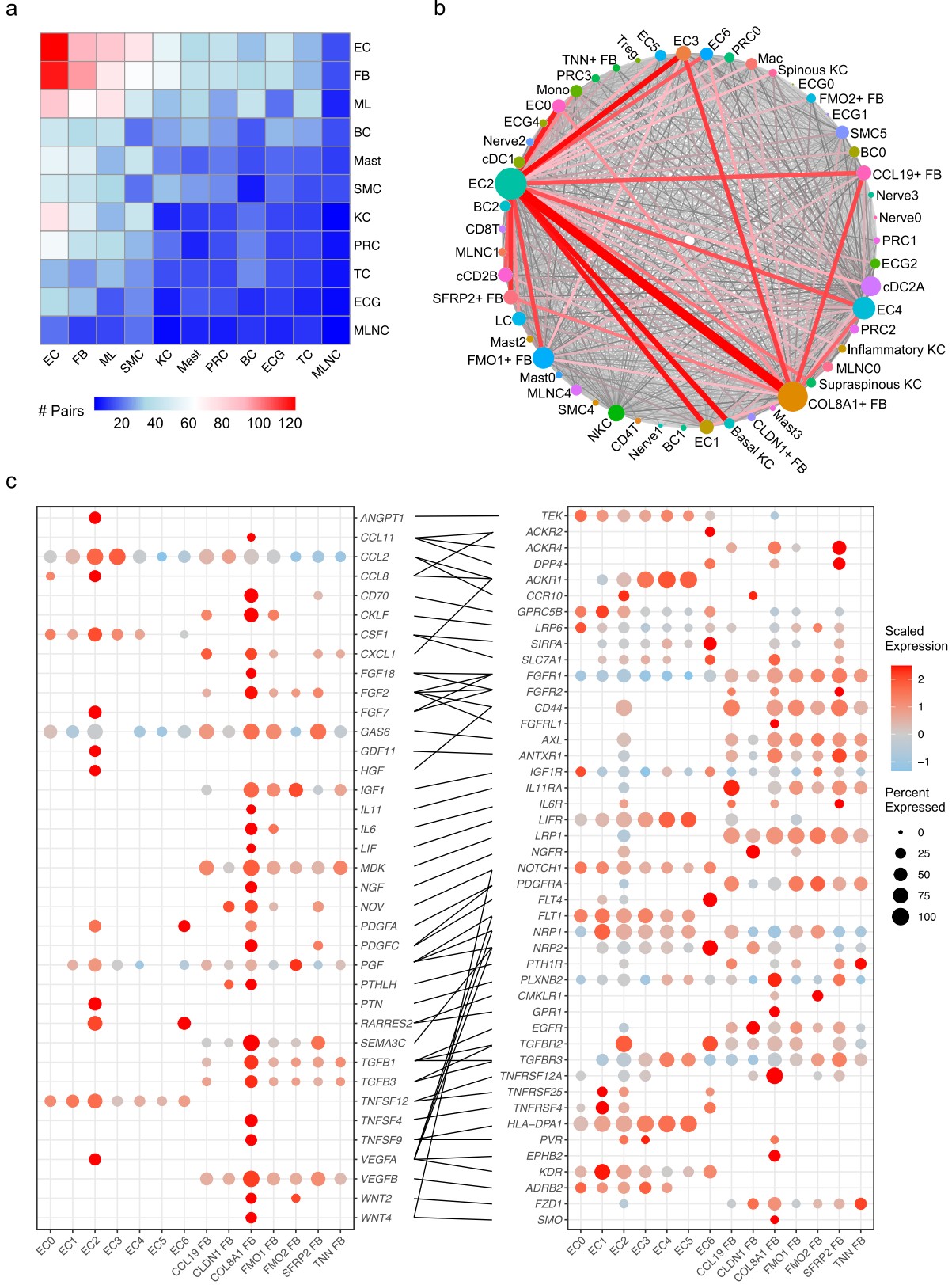

Illumina NovaSeq 6000 sequencer to generate 150 bp paired-end reads. Data processing including quality control, read alignment (hg38), and gene quantification was conducted using the 10X Cell Ranger software. The samples were then merged into a single expression matrix using the cellranger aggr pipeline.

**Cell clustering and cell type annotation**

The R package Seurat (v3.1.2) was used to cluster the cells in the merged matrix. Cells with less than 500 transcripts or 100 genes or more than 1e5 transcripts or 15% of mitochondrial expression were first filtered out as low-quality cells. The NormalizeData function was used

**Fig. 6 | Myofibroblasts and EndoMTs act as central hubs in cell-cell communications. a** Heatmap showing the number of ligand-receptor pairs with interaction scores higher in SSc compared to NS. Row, cell type expressing the ligand; column, cell type expressing the receptor. Color scale, number of ligand-receptor pairs. EC endothelial cell, FB fibroblast, ML myeloid cell, BC B cell. Mast mast cell, SMC smooth muscle cell, KC keratinocyte, PRC pericyte. TC T cell, ECG eccrine gland cell, MLNC melanocyte. **b** Connectome web analysis of interacting subtypes in the SSc samples. Vertex (colored cell node) size is proportional to the number of interactions to and from that cell type, whereas the thickness of the connecting lines is proportional to the number of interactions between two nodes. **c** Dot plots showing expression of the ligands (left) and receptors (right) in endothelial and fibroblast subtypes in the SSc samples. Color scale indicates the level of expression in positive cells, whereas dot size reflects the percentage of cells expressing the gene.

to normalize the expression level for each cell with default parameters. The FindVariableFeatures function was used to select variable genes with default parameters. The ScaleData function was used to scale and center the counts in the dataset. Principal component analysis (PCA) was performed on the variable genes. The RunHarmony function from the Harmony package was applied to remove potential batch effect among samples processed in different batches. Uniform Manifold Approximation and Projection (UMAP) dimensional reduction was performed using the RunUMAP function. The clusters were obtained using the FindNeighbors and FindClusters functions with the resolution set to 0.6. The cluster marker genes were found using the FindAllMarkers function. The cell types were annotated by overlapping the cluster markers with the canonical cell type signature genes. To calculate the disease composition based on cell type, the number of cells for each cell type from each disease condition were counted. The counts were then divided by the total number of cells for each disease condition and scaled to 100 percent for each cell type. Differential expression analysis between any two groups of cells were carried out using the FindMarkers function.

### Cell type sub-clustering
Sub-clustering was performed on the abundant cell types. The same functions described above were used to obtain the sub-clusters. Sub-clusters that were defined exclusively by mitochondrial gene expression, indicating low quality, were removed from further analysis. The subtypes were annotated by overlapping the marker genes for the sub-clusters with the canonical subtype signature genes. The module scores were calculated using the AddModuleScore function on the intended gene lists. The ECM score was calculated on the genes from the extracellular matrix pathway from the Gene Ontology database. The TGF-β and IL-4 score for fibroblast subtypes were calculated on induced genes in fibroblasts after stimulation with TGF-β or IL-4[89]. The module scores for the upstream regulators were calculated on the target gene lists from the Ingenuity Pathway Analysis software.

### Ligand receptor interaction analysis
CellphoneDB (v2.0.0) was applied for ligand-receptor analysis. Each subtype was separated by their disease classifications (SSc or NS), and a separate run was performed for each disease classification. If a subtype contains fewer than 10 cells for a disease classification, it was not considered in the ligand-receptor analysis for this disease classification. Pairs with $p > 0.05$ were filtered out from further analysis. To compare between the two disease conditions, each pair was assigned to the condition in which it showed the higher interaction score. The number of interactions between each subtype pair was then calculated. The connectome web was plotted using the R package igraph.

### Pseudotime trajectory construction
Pseudotime trajectories for the fibroblast and endothelial transitions were constructed using the R package Monocle (v2.22.0). The raw counts for cells were extracted from the Seurat analysis and normalized by the estimateSizeFactors and estimateDispersions functions with the default parameters. Genes detected in >10 cells were retained for further analysis. Variable genes were determined by the differentialGeneTest function with a model against the group identities for the fibroblast transition and the Seurat sub-cluster

identities for the endothelial transition. The orders of the cells were determined by the orderCells function, and the trajectory was constructed by the reduceDimension function with default parameters. Differential expression analysis was carried out using the differentialGeneTest function with a model against the pseudotime, and genes with an adjusted $p$ value smaller than 0.05 were clustered into five patterns and plotted in the heatmap. Ingenuity Pathway Analysis was used to determine the upstream regulators for the genes in each expression pattern. A module score was calculated for each upstream regulator on the target genes from all five patterns. The module scores were calculated using the Seurat function AddModuleScore with default parameters. Pearson correlation was then performed between the upstream regulator module scores and the pseudotime.

### Spatial sequencing library preparation
Skin samples were frozen in OCT medium and stored at −80 °C until sectioning. Optimization of tissue permeabilization was performed on 20 μm sections using Visium Spatial Tissue Optimization Reagents Kit (10X Genomics, Pleasanton, CA, USA), which established an optimal permeabilization time to be 9 min. Samples were mounted onto a Gene Expression slide (10X Genomics), fixed in ice-cold methanol, stained with hematoxylin and eosin, and scanned under a microscope (Keyence, Itasca, IL, USA). Tissue permeabilization was performed to release the poly-A mRNA for capture by the poly(dT) primers that are precoated on the slide and include an Illumina TruSeq Read, spatial barcode, and unique molecular identifier (UMI). Visium Spatial Gene Expression Reagent Kit (10X Genomics) was used for reverse transcription to produce spatially barcoded full-length cDNA and for second strand synthesis followed by denaturation to allow a transfer of the cDNA from the slide into a tube for amplification and library construction. Visium Spatial Single Cell 3′ Gene Expression libraries consisting of Illumina paired-end sequences flanked with P5/P7 were constructed after enzymatic fragmentation, size selection, end repair, A-tailing, adaptor ligation, and PCR. Dual Index Kit TT Set A (10X Genomics) was used to add unique i7 and i5 sample indexes and generate TruSeq Read 1 for sequencing the spatial barcode and UMI and TruSeq Read 2 for sequencing the cDNA insert, respectively. Libraries were then sequenced on the Illumina NovaSeq 6000 sequencer to generate 150 bp paired end reads.

### Spatial sequencing data analysis
After sequencing, the reads were aligned to the human genome (hg38), and the expression matrix was extracted using the spaceranger pipeline. Seurat was then used to analyze the expression matrix. Specifically, the SCTransform function was used to scale the data and find variable genes with default parameters. PCA and UMAP were applied for dimensional reduction. The FindTransferAnchors function was used to find a set of anchors between the spatial-seq data and scRNA-seq data, which were then transferred from the scRNA-seq to the spatial-seq data using the TransferData function. These two functions construct a weight matrix that defines the association between each query cell and each anchor. These weights sum to 1 for each spot and were used as the percentage of the cell type in the spot.

## Immunofluorescence and Immunohistochemistry staining

Formalin-fixed embedded human tissues were sectioned and heated at 65 °C for 30 min, deparaffinized, and rehydrated. Slides were placed in pH 6 or pH 9 antigen retrieval buffer according to manufacture structure and heated at 125 °C for 30 s in a pressure cooker water bath. After cooling, slides were blocked and incubated with primary rabbit and mouse anti-human antibodies. Primary rabbit antibodies were used: anti-alpha smooth muscle actin (αSMA), anti-Col1A1, anti-Vgll3, anti- Vimentin, anti-YAP anti-TEAD1, anti-CD31 antibodies from (Abcam, cat # ab5694, LSB, cat # LS-C343921, Abcam, cat # ab254938, Abcam, cat # ab 92542, Cell Signaling, cat # 12292 and Abcam, cat # ab 32457, respectively). Along with rabbit primary antibodies, appropriate mouse-antihuman antibodies were also used: anti-αSMA (cat # ab 254938), anti-Vimentin (cat # ab 8978), anti-CD31(cat # ab 9498), all from Abcam and TEAD3 antibody from (Abnova, cat # H00007005). All primary rabbit antibodies mentioned above were diluted 1:100, except CD31 1:250 dilution in blocking solution and coincubated with appropriate primary mouse antibodies (α-SMA, Vimentin and TEAD3 1:50 dilutions and CD31 1:100 overnight at 4 °C. Appropriate negative (no primary or secondary antibodies or isotype control antibodies: rabbit IgG (ab172730), mouse IgG1 (ab 280974) both from Abcam, IgG2ak (14-4724-82) from Invitrogen (Supplementary Data 7), antibodies were stained in parallel with each set of the slides mentioned above. Slides were then washed three times for 5 min each with phosphate-buffered saline/ Tween 20 (PBST). All slides were then incubated with secondary antibodies fluorochrome-conjugated Alexa Fluor 594 conjugated anti-rabbit IgG (711-585-152) and Alexa Fluor 488 conjugated anti-mouse IgG (715-545-151) from Jackson Immuno Research. After 30 min coincubation, slides were washed three times for 5 min each with PBST. Mounted in an Prolong Dimond antifade with DAPI (Invitrogen). Photomicrographs were taken on Zeiss fluorescence microscope. All IMF exposures were compared against isotype control. The selected frames used in the figures were representative of the whole biopsy.

Paraffin embedded tissue sections (SSc and control skin) were heated at 60 °C for 30 min, de-paraffinized, and rehydrated. Slides were placed in PH9 antigen retrieval buffer and heated at 125 °C for 30 s in a pressure cooker water bath. After cooling, slides were treated with 3% $H_2O_2$ (5 min) and blocked using 10% goat serum (30 min). Overnight incubation (4 °C) was then performed using anti-human primary antibody. Antibodies used were YAP1 (LifeSpan Biosciences Inc, LS-C331201, pH9, 2ug/ml), WWTR1 (LifeSpan Biosciences Inc, LS-C173295-100, pH9, 1:150), TEAD1, (LifeSpan Biosciences Inc, LS-B3534-100, pH9, 1:100), TEAD3 (LifeSpan Biosciences Inc, LS-C668058, pH9, 1:300), VGLL3 (St. Johns Laboratories, STJ115228-100, pH9, 1:100). Slides were then washed, treated with secondary antibody, peroxidase (30 min) and diaminobenzidine substrate. Counterstain with Hematoxylin and dehydration was done and slides were mounted and viewed under the microscope.

## Isolation and culture of fibroblasts and endothelial cells from SSc skin

Study participants were recruited from the University of Michigan Scleroderma Program. Dermal fibroblasts were isolated from punch biopsies from the distal forearm of healthy volunteers and diffuse cutaneous (dc)SSc patients. All patients met the ACR/EULAR criteria for the classification of SSc[90]. All patients are diagnosed with dcSSc (2 males and 8 females; age 57.5 ± 13.0 years, mean ± SD), and the disease duration was 1.9 ± 0.9 years (mean ± SD). Their skin scores ranged from 0 to 40 with a mean of 19.1 ± 14.2 (mean ± SD). Negatively selected fibroblasts were cultured in RPMI 1640 with 10% FBS and antibiotics. This study was approved by the institutional review board and all participants signed an informed consent prior to enrollment. For endothelial cells, punch biopsies obtained from SSc patients were digested as previously described[91]. After digestion, endothelial cells were magnetically labeled with anti-CD31 antibodies (CD31 MicroBead Kit 130-091-935, Miltenyi Biotech) and purified. These CD31⁺ endothelial cells were maintained in EBM-2 media with growth factors (Lonza CC-3202). Cells between passages 3 and 6 were used in all experiments.

## Cell treatment and transfection

Dermal fibroblasts or endothelial cells from dcSSc patients were treated with 10 μM of LATS kinase inhibitor TRULI/Lats-IN-1 (MedChenExpress HY-138489) or YAP/TEAD inhibitor verteporfin (Cayman Chemical 17334) 0.1–10 μM for 48 to 72 h. Gene knockdown was done using Accell siRNAs (Dharmacon) in dermal fibroblasts and OnTarget siRNAs (Dharmacon) in dermal ECs, following protocols recommended by the manufacturer.

## Functional endpoint experiments

Gene expression changes in cells were performed by qPCR after total RNA was extracted using Direct-zol RNA MiniPrep Kit (Zymo Research R2052). Quantitative PCR was performed in a ViiA 7 Real-Time PCR System. Protein expression changes was monitored using Western blotting. After blocking, the blots were probed with antibodies against collagen I (COL1, Abcam ab6308), VWF (Novus NBP2-33003) or αSMA (Abcam ab5694). For loading control, the blots were immunoblotted with antibodies against GAPDH (Cell Signaling #2118). Band quantification was performed using ImageJ. The IncuCyte Live-Cell Imaging System was used to monitor cell proliferation or migration. After adding different treatments cells were monitored by IncuCyte. Cell counts were analyzed by the IncuCyte S3 Analysis software. Gel contraction assays was performed using the cell contraction kit from Cell Biolabs (CBA-201). Immunofluorescence on cells was performed using anti-SMA antibodies (Abcam ab5694) or anti-VWF antibodies (Novus NBP2-33003) followed by Alexa Fluor secondary antibodies (Thermo Fisher).

## Statistical analysis for in vitro experiments

For the in vitro experiments, normality test was first conducted to determine whether the data is normally distributed or skewed. To determine the differences between groups, unpaired $t$ test, Mann–Whitney $U$ test, one-way ANOVA with Sidak test, Kruskal–Wallis test with Dunn's test, or two-way ANOVA with Dunnett's test were performed using GraphPad Prism version 8 (GraphPad Software, Inc). $P$ values of less than 0.05 were considered statistically significant. Results were expressed as mean ± SD unless specified.

## Reporting summary

Further information on research design is available in the Nature Portfolio Reporting Summary linked to this article.

## Data availability

The single cell RNA-seq and spatial-seq data generated in this study have been deposited in the GEO database under accession code GSE249279. All other data are available in the article and its Supplementary files or from the corresponding author upon request. Source data are provided with this paper.

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

## Acknowledgements

This work was supported by the Taubman Institute via Innovative Program (J.M.K., J.E.G.), Parfet Emerging Scholar (J.M.K.), and

Taubman Emerging Scholar (A.C.B.) funds; the Babcock Endowment Fund (L.C.T., J.E.G.); and the National Institute of Health (NIH): K08-AR078251 (A.C.B.), R01-AR071384 (J.M.K.), K24-AR076975 (J.M.K.), R01-AR069071 (J.E.G.), R01-AR073196 (J.E.G.), P30-AR075043 (J.E.G.), R21 AR077741 (J.E.G.), K01-AR072129 (L.C.T.), R01-AI022553 (R.L.M.), R01-AR040312 (R.L.M.), R01-AR074302 (R.L.M.), K24-AR063120 (D.K.), University of Michigan Clinical Autoimmunity Center of Excellence 1UM1AI144298 (D.K., J.M.K.) and the Lupus Research Alliance (J.M.K.). The Department of Defense (P.T.), the Beverley and Michael Townsley Fund for Scleroderma Biorepository (D.K. and P.T.), the Edward T. and Ellen K. Dryer Early Career Professor of Rheumatology (P.T.). The Dermatology Foundation (A.C.B., L.C.T. and J.E.G.). Arthritis National Research Foundation (L.C.T.).

## Author contributions

Conceptualization: F.M., D.K., and J.E.G.; Methodology: F.M., P.T., M.G.K., O.P., M.P., and J.E.G.; Imaging: X.X. and O.P.; Investigation: F.M., P.T., J.K., R.W., G.A.H., P.W.H., Y.J., E.X., M.N., D.O., W.D.B., S.P., E.M., M.P., R.L.M., J.V., L.C.T., R.L., J.M.K., and A.C.B.; Writing: F.M., P.T., and J.E.G.; Funding Acquisition: D.K. and J.E.G.; Supervision: M.P., R.L.M., D.K., and J.E.G.

## Competing interests

J.E.G. has received Grant support from Celgene/BMS, Janssen, Eli Lilly, and Almirall. J.E.G. has served on advisory boards for AstraZeneca, Sanofi, Eli Lilly, Boehringer Ingelheim, Novartis, Janssen, Almirall, BMS. J.M.K. has received Grant support from Q32 Bio, Celgene/BMS, Ventus Therapeutics, and Janssen. J.M.K. has served on advisory boards for AstraZeneca, Eli Lilly, GlaxoSmithKline, Bristol Myers Squibb, Avion Pharmaceuticals, Provention Bio, Aurinia Pharmaceuticals, Ventus Therapeutics, Vera Therapeutics, and Boehringer Ingelheim. P.W.H. has received effort support from Q32 Bio. The remaining authors declare no competing interests.
