## [Peer Review File · Nature Communications]

Systems-based identification of the Hippo pathway for promoting fibrotic mesenchymal differentiation in systemic sclerosisEditorial Note: Parts of this Peer Review File have been redacted as indicated to maintain the confidentiality of unpublished data.

REVIEWER COMMENTS

Reviewer #1 (Remarks to the Author):

Positives-This is a high quality paper from an expert group. The paper addresses an important area of unmet need: understanding the biomechanisms underlying fibrosis in systemic sclerosis. The methodology is comprehensive and has some novelty.

Reservations-The main findings, namely involvement of myofibroblasts and myofibroblast precursors in the fibrotic process, do not come as a surprise, but are consistent with multiple published data. Also, the relevance of the Hippo pathway and EndoMT to this process, have also been demonstrated with simpler hypothesis driven approaches.

Requests for revision/clarification-lines 492 to 503 under Methodology. It appears that differing methodologies were used at each Centre for the single cell RNAseq library preparation. This is not recommended and it would be important to include justification of this and also to detail how many of the SSc and healthy control samples were prepared by each method at the respective Centres.

Reviewer #2 (Remarks to the Author):

This study is based on scRNA sequencing of 22 biopsies obtained from patients with systemic sclerosis (SSc) and corresponding controls, and linking these data to results from spatial transcriptomics. Clusters and subpopulations of different cell types are identified, the functional relevance of a defined fibroblast subpopulation (COL8 pos.) and of an endothelial-to-mesenchymal transitioning cell population for ECM production has been delineated, which show considerable overlap in their signatures, and the importance of Hippo signaling is shown by enhancing as well as downmodulating the pathway.

The type of study with combined use of scRNA seq and spatial transcriptome analysis, analyzed by state-of-the-art data processing, is to my knowledge the first in the field, therefore novel and it is of high quality. It provides a wealth of novel data that is clearly structured and discussed in a well-organized manner by experts in the field. The results deriving from data processing combined with cell-based studies (primary cells isolated from patient and control biopsies) raise highly relevant concepts that are expected to strongly impact research in the fibrosis field.

Some points to discuss:

1. I recommend to reconsider the title of the manuscript, what is the evidence for 'maintenance' of the fibrotic differentiation. A stronger emphasis on that point in discussion/results will also help.
2. The finding that mechanosignaling and modulation by verteporfin plays an important role in the formation of myofibroblasts is not novel, as the authors also clearly point out. However, do they have an explanation for the lack of YAP signals in 2 of 4 SSc samples?
3. Their concept of a differentiation pathway from SFRP2+ to COL8+ fibroblasts is interesting. Since these different subtypes are located in the upper/lower (respectively) compartment of the dermis, it would be interesting to know if there is a mixed population expressing both transcripts in the space in between. Incidentally, have the authors detected COL8 protein produced by SSc fibroblasts?
4. The authors provide data in the supplementary tables on the 2 FB subtypes (my point 3), but mention several more. Are those data available upon request or did I miss them?
5. The authors mention 'the intriguing possibility that VGLL3 may contribute to the female bias in SSc ...'. Please elaborate how this can be explained.
6. Minor point: The figure legends are very brief and would benefit from more detail.

Reviewer #3 (Remarks to the Author):

The objective of the study titled "Hippo effectors promote and maintain fibrotic mesenchymal differentiation in systemic sclerosis" is to offer a detailed understanding of the cellular composition and cell-to-cell communication in SSc skin. The study highlights specific subsets of myofibroblast differentiation and endothelial-to-mesenchymal transition (EndoMT) as the primary contributors to fibrosis, emphasizing the significant role of Hippo pathway regulation in promoting a pro-fibrotic phenotype in both myofibroblasts and EndoMTs. Although the involvement of myofibroblasts in the development of SSc skin fibrosis has been extensively studied, the authors aimed to enhance this knowledge by providing spatial context, comprehensive cell trajectories, and signaling mechanisms to further characterize myofibroblast differentiation. Furthermore, the authors sought to define the role of EndoMT in SSc skin fibrosis, considering its implication in fibrosis pathogenesis in other fibrotic conditions. As a result, they discovered that inhibiting TEAD transcriptional activity downstream in the Hippo pathway significantly mitigates the pro-fibrotic phenotype in both fibroblast and endothelial cells in SSc skin. This manuscript's premise is significant as it aims to provide a comprehensive understanding of myofibroblast differentiation in SSc skin, which has not yet been achieved. By doing so, the authors have advanced the field of SSc by identifying potential new treatment targets and highlighting the importance of EndoMT and the Hippo pathway in fibrosis development. The scientific work presented in the manuscript is impressively comprehensive and utilizes various appropriate techniques to support the findings. However, there are several issues that the authors need to address before the manuscript is ready for publication.

Major Issues

- In both the abstract and discussion (lines 40 and 449, respectively), the authors state that modulation of the Hippo pathway can reverse the pro-fibrotic phenotypes in myofibroblasts and EndoMTs. While the data presented is convincing that fibrosis is attenuated, without a direct comparison to normal skin fibroblasts and endothelial cells, they have not demonstrated that this is reversed. They should provide data for non-diseased cells for Figures 3 and 5 to enhance their conclusion.
- One big concern here is an overreliance on methods like CellChat and CellPhone DB, which identify communication pathways but they are identified bioinformatically and don't

necessarily correlate with expression levels. For example, Supp. Fig. 3b, CYR61 and VGLL3 are barely if at all higher expressed in group 3 compared to group 2. A larger analysis of what may be the most relevant of communication pathways and validation by another spatial method would be useful to distill those signals that may be the most likely drugable targets.

- For Fig. 3k, the third panel indicates there is no significant increase in COL1 under TRULI compared to NT, as there's no asterisk above the TRULI bar in the 2nd graph. If this is the case, it is misleading to state "similar results were found at protein level" (line 220) compared to the "significant increase in...COLA1" (line 218, 219) mRNA in panel K.

Minor Issues

- Lines 79 and 80: grammar issue, not complete sentence, but also the phrasing is a little too strong for the given references which are not definitive in the respective roles of cytotoxic T cells and anti-endothelial autoantibodies.
- Line 125: Please provide explanation for why number of fibroblasts is not increased in the SSc samples.
- Line 132: Make clear these samples are a separate set of 4 SSc patients specifically for spatial.
- Lines 154-156: Reference 48 does not mention any of the 7 marker genes listed here and reference 47 mentions only SFRP2. COL8A1 is extremely important to the manuscript's exciting results, please provide references appropriate for these markers.
- Line 157: for clarification and reader understanding, make explicit ACTA2, etc are myofibroblast genes and provide references.
- Line 215, 217: Provide references for TRULI and verteporfin and their reported mechanism of action.
- In Fig. 4j, why is there increased CD31 fluorescence in SSc skin compared to NS, if endothelial apoptosis is a known event in SSc?
- In Fig. 5b, add in quantifications of the western blots, like in Fig. 3k.
- In Fig. 5c, why are TRULI treated cells so less confluent than NT and verteporfin? Shouldn't increased nuclear YAP cause increased proliferation? Address this.
- Line 288: Its not addressed why, if loss of VWF is due to increased EndoMT, PECAM is increased with TRULI treatment in Fig. 5a for presumably the same treatment duration of

48-72 hrs.

- Fig. 5e, 4th graph, knockdown of VGLL3 and TEAD3 increasing CDH5 is not consistent with Fig. 5a, where CDH5 had a significant decrease with verteporfin. This echoes the discrepancy in panel a between PECAM and CDH5, and should be addressed in the Discussion.
- Line 293: Need to add in “knock-down” in-between “VGLL3” and “appeared”, otherwise it is confusing.
- Line 368: For clarity and ease of comprehension for the reader, authors should mention GJA4’s enrichment in cluster 0 in the Results section for Fig. 4, as it was confusing to encounter this result for the first time in the discussion.
- Line 426: its is unclear why the authors are making a connection between VGLL3 and SSc female bias. Please provide more exposition and references as appropriate.
- Line 437, 438: Which figure actually visualizes EndoMT in upper & mid-dermis? Fig. 5 is cell culture and Fig. 4 does not convincingly show decreased endothelial markers and increased fibrosis in upper & mid-dermis.
- Methods: Human sample acquisition, please provide the location of the body biopsies were taken from, if this location was consistent, if biopsies were from active lesions, age range and gender distribution, and disease duration.
- Add “n= ___” for all relevant figure legends.

REVIEWER COMMENTS

Reviewer #1 (Remarks to the Author):

Positives-This is a high quality paper from an expert group. The paper addresses an important area of unmet need: understanding the biomechanisms underlying fibrosis in systemic sclerosis. The methodology is comprehensive and has some novelty.

A: Thank you for those positive comments.

Reservations-The main findings, namely involvement of myofibroblasts and myofibroblast precursors in the fibrotic process, do not come as a surprise, but are consistent with multiple published data. Also, the relevance of the Hippo pathway and EndoMT to this process, have also been demonstrated with simpler hypothesis driven approaches.

A: We believe the strength of our data lies in the unbiased approach through single-cell RNA-sequencing. While the myofibroblasts have been reported on, and which papers we cite in our manuscript, we believe the endo-MT pathway that we report to be novel.

Requests for revision/clarification-lines 492 to 503 under Methodology. It appears that differing methodologies were used at each Centre for the single cell RNAseq library preparation. This is not recommended and it would be important to include justification of this and also to detail how many of the SSc and healthy control samples were prepared by each method at the respective Centres.

A: We apologize, but this was not entirely accurate, the epidermal and dermis from the samples at Michigan were digested separately but then combined in a 1:1 ratio before library preparation. Only 3 of the Michigan samples had epidermis and dermis library preparation done separately and we did not observe that those samples were any different than the ones where the epidermis and dermis were combined. We have corrected this in our revised application. We'd like to stress that it is routine to combine different datasets for single-cell analyses. All the data used here were done on the 10X Chromium System using the V3 library preparation. All files were processed from raw data (Fastq files) and batch-corrected using Harmony. We used rigorous approaches to ensure that there were no batch effects or overriding individual effects in our dataset., as is described in the methods.

Reviewer #2 (Remarks to the Author):

This study is based on scRNA sequencing of 22 biopsies obtained from patients with systemic sclerosis (SSc) and corresponding controls, and linking these data to results from spatial transcriptomics. Clusters and subpopulations of different cell types are identified, the functional relevance of a defined fibroblast subpopulation (COL8 pos.) and of an endothelial-to-mesenchymal transitioning cell population for ECM production has been delineated, which show considerable overlap in their signatures, and the importance of Hippo signaling is shown by enhancing as well as downmodulating the pathway.

The type of study with combined use of scRNA seq and spatial transcriptome analysis, analyzed by state-of-the-art data processing, is to my knowledge the first in the field, therefore novel and it is of high quality. It provides a wealth of novel data that is clearly structured and discussed in a well-organized manner by experts in the field. The results deriving from data processing combined with cell-based studies (primary cells isolated from patient and control biopsies) raise highly relevant concepts that are expected to strongly impact research in the fibrosis field.

A: Thank you for those thoughtful and positive comments.

Some points to discuss:

1. I recommend to reconsider the title of the manuscript, what is the evidence for 'maintenance' of the fibrotic

differentiation. A stronger emphasis on that point in discussion/results will also help.

A: We acknowledge the point made by the reviewer and agree with it. We have removed “maintain” from the title and it currently reads “*Hippo effectors promote fibrotic mesenchymal differentiation in systemic sclerosis*”.

2. The finding that mechanosignaling and modulation by verteporfin plays an important role in the formation of myofibroblasts is not novel, as the authors also clearly point out. However, do they have an explanation for the lack of YAP signals in 2 of 4 SSc samples?

A: We are not sure which data the reviewer is referring to. Overall, the results across all samples for both single-cell analyses and the *in vitro* experiments were consistent across all samples with significant differences between the groups.

3. Their concept of a differentiation pathway from SFRP2+ to COL8+ fibroblasts is interesting. Since these different subtypes are located in the upper/lower (respectively) compartment of the dermis, it would be interesting to know if there is a mixed population expressing both transcripts in the space in between. Incidentally, have the authors detected COL8 protein produced by SSc fibroblasts?

A: We thank the reviewer for this suggestion. We first plotted *COL8A1* gene expression in the two spatial samples where myofibroblasts were captured. However, due to limited sensitivity of the Visium technique, *COL8A1* gene was detected in only a few spots in each sample, although most of these spots were in deeper dermis. To determine whether there is a population undergoing the transition from *SFRP2+* FB to *COL8A1+* FB, we generated scatter plots showing the *SFRP2+* FB score and *COL8A1+* FB score in the fibroblast-rich spots (as shown in Fig. 2f) within these two samples. Based on the scores derived from the analysis, we identified a subset of spots that displayed intermediate scores for both subtypes, which likely correspond to regions in the transitional state.

[Figure redacted]

4. The authors provide data in the supplementary tables on the 2 FB subtypes (my point 3), but mention several more. Are those data available upon request or did I miss them?

A: We identified 7 major fibroblast subtypes in healthy and SSc skin and listed the top four marker genes for each in Figure 2E. We will release all data related to this manuscript upon publication in a public domain, and the Seurat object will be available upon request.

5. The authors mention 'the intriguing possibility that *VGLL3* may contribute to the female bias in SSc ...'. Please elaborate how this can be explained.

A: Thank you. We have expanded on this and provided additional background. We have added “*VGLL3 has been previously implicated as a promoter of sex-biased autoimmune responses (PMID: 27992404), and VGLL3 has recently been implicated in cardiac fibrosis promoting myofibroblast collagen production (PMID:36754961). Our data are consistent with such a role for VGLL3 both in relation to myofibroblast function and collagen I expression, as well as the development of Endo-MTs*”

6. Minor point: The figure legends are very brief and would benefit from more detail.

A: We have added more details into the figure legends of selected figures to make them clearer, particularly the number of replicates, etc.

Reviewer #3 (Remarks to the Author):

The objective of the study titled "Hippo effectors promote and maintain fibrotic mesenchymal differentiation in systemic sclerosis" is to offer a detailed understanding of the cellular composition and cell-to-cell communication in SSc skin. The study highlights specific subsets of myofibroblast differentiation and endothelial-to-mesenchymal transition (EndoMT) as the primary contributors to fibrosis, emphasizing the

significant role of Hippo pathway regulation in promoting a pro-fibrotic phenotype in both myofibroblasts and EndoMTs. Although the involvement of myofibroblasts in the development of SSc skin fibrosis has been extensively studied, the authors aimed to enhance this knowledge by providing spatial context, comprehensive cell trajectories, and signaling mechanisms to further characterize myofibroblast differentiation. Furthermore, the authors sought to define the role of EndoMT in SSc skin fibrosis, considering its implication in fibrosis pathogenesis in other fibrotic conditions. As a result, they discovered that inhibiting TEAD transcriptional activity downstream in the Hippo pathway significantly mitigates the pro-fibrotic phenotype in both fibroblast and endothelial cells in SSc skin. This manuscript's premise is significant as it aims to provide a comprehensive understanding of myofibroblast differentiation in SSc skin, which has not yet been achieved. By doing so, the authors have advanced the field of SSc by identifying potential new treatment targets and highlighting the importance of EndoMT and the Hippo pathway in fibrosis development. The scientific work presented in the manuscript is impressively comprehensive and utilizes various appropriate techniques to support the findings. However, there are several issues that the authors need to address before the manuscript is ready for publication.

A: Thank you for those thoughtful and positive comments. We have responded to each of the critiques below and hope that it is to the reviewer's satisfaction.

Major Issues

- In both the abstract and discussion (lines 40 and 449, respectively), the authors state that modulation of the Hippo pathway can reverse the pro-fibrotic phenotypes in myofibroblasts and EndoMTs. While the data presented is convincing that fibrosis is attenuated, without a direct comparison to normal skin fibroblasts and endothelial cells, they have not demonstrated that this is reversed. They should provide data for non-diseased cells for Figures 3 and 5 to enhance their conclusion.

A: We thank the reviewer for this comment. We have now added data generated from fibroblasts or endothelial cells isolated from healthy controls in both fibroblasts (Supplemental Figure 3) and ECs (Figure 5). The results section has been updated accordingly.

- One big concern here is an overreliance on methods like CellChat and CellPhone DB, which identify communication pathways but they are identified bioinformatically and don't necessarily correlate with expression levels. For example, Supp. Fig. 3b, CYR61 and VGLL3 are barely if at all higher expressed in group 3 compared to group 2. A larger analysis of what may be the most relevant of communication pathways and validation by another spatial method would be useful to distill those signals that may be the most likely druggable targets.

A: We agree with the reviewer in regard to this comment. The receptor-ligand interactions that are highlighted by bioinformatic tools such as CellChat and CellPhoneDB are based on established ligand-receptor interactions but lack spatial context, which we agree with the reviewer is necessary for a more complete biological interpretation. However, using spatial data to predict or validate receptor-ligand interactions is still in its infancy, and related to that, a major limitation is that the resolution of the Visium system is in the range of 55um; another limitation is that the sensitivity of the spatial-sequencing technologies in regard to gene expression detection is substantially lower than that of single-cell sequencing. Therefore, the absence of receptor-ligand interactions in a spatial context would not necessarily mean that it is not there, and we feel that shifting the focus towards this in the spatial context in our manuscript could be distracting and also not be feasible given the potential number of such receptor-ligand pairs. We agree with the reviewer on the CYR61 and VGLL3 – the changes may be subtle but correlate with the pseudotime trajectory across those three groups – CYR61 was selected as a marker gene for the Hippo signaling pathway as a whole. As shown in Figure 3i, the enrichment for the correlation of the Hippo-signaling pathway across the three groups is in the $p < 10^{-100}$ range for significance.

- For Fig. 3k, the third panel indicates there is no significant increase in COL1 under TRULI compared to NT, as there's no asterisk above the TRULI bar in the 2nd graph. If this is the case, it is misleading to state "similar results were found at protein level" (line 220) compared to the "significant increase in...COLA1" (line 218, 219) mRNA in panel K.

A: Thank you for catching that. We have revised this in our results and now state: “We validated this for SMA with western blotting, but the increase in COL1A1 did not reach significance (**Fig. 3k**), these findings were validated by immunofluorescent staining with increased expression of SMA but only slight to moderate increase in COL1A1 expression (**Fig. 3l**).” We hope that the reviewer is satisfied with these changes.

Minor Issues

- Lines 79 and 80: grammar issue, not complete sentence, but also the phrasing is a little too strong for the given references which are not definitive in the respective roles of cytotoxic T cells and anti-endothelial autoantibodies.

A: We apologize, but we're not sure what “grammar issue” the reviewer is referring to, and this being an incomplete sentence. We qualify the statement towards the “endothelial damage” as that it “may” be “immune-mediated”. We're unsure how this would be seen as too strong or forceful of a statement. Furthermore, with those references, we're not emphasizing those as factual but as evidence for this as a potential mechanism.

- Line 125: Please provide explanation for why number of fibroblasts is not increased in the SSc samples.

A: We respectfully feel that this would not add much to the results of our findings as we show the proportion of cells in Figure 1c. As can be seen from the figure, there is a very slight increase in fibroblast proportion in scleroderma, but explaining this would be speculative and not fall within the “results” section of our manuscript. In addition, we feel that focusing on the number of fibroblasts (with the exception of myofibroblasts which are not found in normal skin) would not provide much insight as it is the fibroblast subsets and transcriptional shifts that are more relevant for the disease pathogenesis as we show in the data shown in figure 3.

- Line 132: Make clear these samples are a separate set of 4 SSc patients specifically for spatial.

A: Thank you, we have added this to our revised manuscript.

- Lines 154-156: Reference 48 does not mention any of the 7 marker genes listed here and reference 47 mentions only SFRP2. COL8A1 is extremely important to the manuscript's exciting results, please provide references appropriate for these markers.

A: Thank you for noticing that. We have provided several references, mostly single-cell based, to provide references appropriate for these markers.

- Line 157: for clarification and reader understanding, make explicit ACTA2, etc are myofibroblast genes and provide references.

A: Thank you. We have added a reference to demonstrate better that the genes we identify in the COL8A1+ subsets, including ACTA2, are known myofibroblast markers.

- Line 215, 217: Provide references for TRUL1 and verteporfin and their reported mechanism of action.

A: Thank you. We have added references for the function of both TRUL1 and Verteporfin.

- In Fig. 4j, why is there increased CD31 fluorescence in SSc skin compared to NS, if endothelial apoptosis is a known event in SSc?

A: That's a great question and we're not sure precisely the reason. As stated in our manuscript, vascular changes are an early event in systemic sclerosis, as manifested in Raynaud's syndrome, and altered capillary architecture. While reduced vascular density is described, some of the changes are manifested in dilated capillary loops, as can be seen in the periungual area of patients with systemic sclerosis. While our biopsies were not from the periungual area, we can speculate that these altered capillary structures with ACTA2 expression may reflect some dilated capillary loops. As seen in the IF below (ACTA2=red, CD31=green), this was a consistent feature in all the samples we obtained from our subjects.

[Figure redacted]

- In Fig. 5b, add in quantifications of the western blots, like in Fig. 3k.

A: Quantification of the Western blots were added.

- In Fig. 5c, why are TRULI treated cells so less confluent than NT and verteporfin? Shouldn't increased nuclear YAP cause increased proliferation? Address this.

A: The reviewer made a great point regarding the effect of TRULI on cell proliferation. We now replaced the figures with better images. We also included staining for healthy ECs with the treatments as well.

- Line 288: Its not addressed why, if loss of VWF is due to increased EndoMT, PECAM is increased with TRULI treatment in Fig. 5a for presumably the same treatment duration of 48-72 hrs.

A: In Figure 5a, at the RNA level, we showed that TRULI induced ACTA2, and COL1A1 expression significantly in SSc ECs, while downregulated PECAM1 significantly, and CDH5 remained unchanged (no statistical significance was observed). We also confirmed the impact of TRULI in inducing EndoMT by immunofluorescence and Western blotting, looking at the protein levels and morphological changes. The discordance between transcriptomic data and protein expression is not uncommon. We speculate this could be due to posttranscriptional and/or translational modifications or different RNA and protein degradation rates. We added this to the results.

- Fig. 5e, 4th graph, knockdown of VGLL3 and TEAD3 increasing CDH5 is not consistent with Fig. 5a, where CDH5 had a significant decrease with verteporfin. This echoes the discrepancy in panel a between PECAM and CDH5, and should be addressed in the Discussion.

A: The discrepancy between the verteporfin and the knockdown results could be related to the off-target effects of the chemical. It is also possible that TEAD3 and VGLL3 play a more critical role in the Hippo pathway in mediating the EndoMT phenotype than YAP alone. We have plans to explore the individual roles of each of these mediators in future studies. This is added to the results section, where Figure 5 is presented.

- Line 293: Need to add in "knock-down" in-between "VGLL3" and "appeared", otherwise it is confusing.

A: Thank you for noticing this – we have corrected this.

- Line 368: For clarity and ease of comprehension for the reader, authors should mention GJA4's enrichment in cluster 0 in the Results section for Fig. 4, as it was confusing to encounter this result for the first time in the discussion.

A: Thank you for noticing this – we have added a clarification in the results to address this for GJA4.

- Line 426: its is unclear why the authors are making a connection between VGLL3 and SSc female bias. Please provide more exposition and references as appropriate.

A: Thank you. As in our response to reviewer #2, we have expanded on this and provided additional background. We have added, "*VGLL3 has been previously implicated as a promoter of sex-biased autoimmune responses (PMID: 27992404), and VGLL3 has recently been implicated in cardiac fibrosis promoting myofibroblast collagen production (PMID:36754961). Our data are consistent with such a role for VGLL3 both in relation to myofibroblast function and collagen I expression, as well as the development of Endo-MTs*"

- Line 437, 438: Which figure actually visualizes EndoMT in upper & mid-dermis? Fig. 5 is cell culture and Fig. 4 does not convincingly show decreased endothelial markers and increased fibrosis in upper & mid-dermis.

A: The EndoMT would be represented by CD31+SMA+ markers shown in Figure 4G. To make this clearer we have revised this sentence, which now states “*These early changes coincide with the same dermal locations where EndoMTs, characterized by SMA+CD31+, are prominent*”

- Methods: Human sample acquisition, please provide the location of the body biopsies were taken from, if this location was consistent, if biopsies were from active lesions, age range and gender distribution, and disease duration.

A: Thank you. We have added this to our Human Sample description. Patient demographic and other information, including age, disease duration, mRSS, FVC, DLCO, treatment, and sex, is provided in Supplemental Table 1.

- Add “n= ___” for all relevant figure legends.

A: Thank you. This has been added to our revised manuscript.

REVIEWERS' COMMENTS

Reviewer #1 (Remarks to the Author):

Thank you for your responses and comments, which have addressed the questions raised.

I am happy with the responses. The article is comprehensive, very high quality and adds to the field and should be accepted in the current revised form.

Reviewer #2 (Remarks to the Author):

The authors have addressed my concerns to my satisfaction.

Reviewer #3 (Remarks to the Author):

We feel that the author made a good faith effort to address the comments of the reviewer in the manuscript, and have addressed most of the criticisms satisfactorily. However, would still say that some comment in the manuscript should be made about the CellChat and CellPhone DB that these networks remain to be validated in future studies with developing technologies. It would be a bit better to put the data out there with the qualifier that these observations have not necessarily been validated.

REVIEWER COMMENTS

Reviewer #3 (Remarks to the Author):

We feel that the author made a good faith effort to address the comments of the reviewer in the manuscript, and have addressed most of the criticisms satisfactorily. However, would still say that some comment in the manuscript should be made about the CellChat and CellPhone DB that these networks remain to be validated in future studies with developing technologies. It would be a bit better to put the data out there with the qualifier that these observations have not necessarily been validated.

Response: We added sentence “Despite previously reported interactions, many ligand-receptor interactions inferred from gene expression require experimental validation in future studies” in Discussion to address the reviewer’s concern.